# ICAM: RETHINKING INSTANCE-CONDITIONED ADAPTATION IN NEURAL VEHICLE ROUTING SOLVER

## ABSTRACT

The neural combinatorial optimization (NCO) method has shown great potential for solving routing problems without requiring expert knowledge. However, existing constructive NCO methods still struggle to solve large-scale instances, which significantly limits their application prospects. To address these crucial shortcomings, this work proposes a novel Instance-Conditioned Adaptation Model (ICAM) for better large-scale generalization of neural routing solvers. In particular, we design a simple yet efficient instance-conditioned adaptation function to significantly improve the generalization performance of existing NCO models with a small time and memory overhead. In addition, with a systematic investigation on the performance of information incorporation between different attention mechanisms, we further propose a powerful yet low-complexity instance-conditioned adaptation module to generate better solutions for instances across different scales. Experimental results show that our proposed method is capable of obtaining promising results with a very fast inference time in solving Traveling Salesman Problems (TSPs), Capacitated Vehicle Routing Problems (CVRPs) and Asymmetric Traveling Salesman Problems (ATSPs). To the best of our knowledge, our model achieves state-of-the-art performance among all RL-based constructive methods for TSPs and ATSPs with up to 1,000 nodes and extends state-of-the-art performance to 5,000 nodes on CVRP instances, and our method also generalizes well to solve cross-distribution instances.

## 1 INTRODUCTION

The Vehicle Routing Problem (VRP) plays a crucial role in various logistics and delivery applications, where the solution quality directly affects the transportation cost and service efficiency (Tiwari & Sharma, 2023; Sar & Ghadimi, 2023). However, efficiently solving VRPs is a challenging task due to their NP-hard nature. Over the past few decades, extensive heuristic algorithms, such as LKH3 (Helsgaun, 2017) and HGS (Vidal, 2022), have been proposed to address different VRP variants. Although these approaches have shown promising results for specific problems, the algorithm designs heavily rely on expert knowledge and a deep understanding of each problem. Moreover, the runtime required for a heuristic algorithm often increases exponentially as the problem scale grows. These limitations greatly hinder the practical application of classical heuristic algorithms.

Over the past few years, different neural combinatorial optimization (NCO) methods have been explored to solve various routing problems (Li et al., 2022; Bengio et al., 2021). In this work, we focus on the constructive NCO method (also known as the end-to-end method) that builds a learning-based model to directly construct an approximate solution for a given instance without any expert knowledge (Vinyals et al., 2015; Kool et al., 2019; Kwon et al., 2020). These methods usually have a faster runtime compared to classical heuristic algorithms, making them a desirable choice to tackle real-world problems with real-time requirements. Existing constructive NCO methods can be divided into two categories: supervised learning (SL)-based (Vinyals et al., 2015; Xiao et al., 2024) and reinforcement learning (RL)-based ones (Nazari et al., 2018; Bello et al., 2016). The SL-based method requires a lot of problem instances with labels (i.e., the optimal solutions of these instances) as its training data. However, it could be extremely hard to obtain sufficient optimal solutions for some complex problems, which impedes its practicality. RL-based methods can learn NCO models by repeatedly interacting with the environment without any labeled data. Nevertheless, due to the

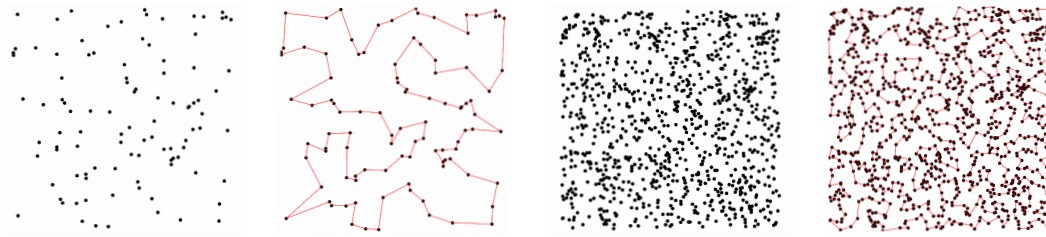

(a) TSP instance with 100 nodes.  (b) TSP instance with 1,000 nodes.

Figure 1: Comparison of two TSP instances and their optimal solutions with different scales (Left: Instance, Right: Solution). The patterns and geometric structures are quite different for these instances. In this work, we propose a powerful Instance-Conditioned Adaptation Model (ICAM) to leverage these instance-specific patterns to directly generate promising solutions for instances across quite different scales.

high memory and computational overhead, it is unrealistic to train the RL-based NCO model directly on large-scale problem instances.

Current RL-based NCO methods typically train the model on small-scale instances (e.g., with 100 nodes) (Kool et al., 2019; Kwon et al., 2020) and then generalize it to tackle larger-scale instances (e.g., with 1,000 nodes). Although these models demonstrate good performance on instances of similar scales to the ones they were trained on, they struggle to generate reasonable good solutions for instances with much larger scales. Recently, two different types of attempts have been explored to address the crucial limitation of RL-based NCO on large-scale generalization. The first one is to perform an extra search procedure on model inference to improve the quality of solution over greedy generation Hottung et al. (2022); Choo et al. (2022). However, this approach typically requires expert-designed search strategies and can be time-consuming when dealing with large-scale problems. The second approach is to train the model on instances of varying scales Khalil et al. (2017); Cao et al. (2021); Zhou et al. (2023). However, learning cross-scale features effectively for better generalization performance remains a key challenge for NCO methods.

In solving routing problems, some recent works reveal that incorporating auxiliary information (e.g., node-to-node distances) in training can improve the model's convergence efficiency and final performance (Son et al., 2023; Jin et al., 2023; Li et al., 2023a; Gao et al., 2024; Wang et al., 2024). However, regarding the information incorporation strategy, existing methods either simply utilize the node-to-node distances to bias the output score in the decoding phase (Son et al., 2023; Jin et al., 2023; Wang et al., 2024) or refine the information via a complex policy (Li et al., 2023a; Gao et al., 2024). Some recent methods, such as ELG (Gao et al., 2024) and DAR (Wang et al., 2024), have shown good performance on large-scale routing instances. However, for routing instances with different scales, the general RL-based methods cannot truly capture instance-specific features according to the changes in geometric structures, which results in still unsatisfactory generalization performance.

In this work, we propose a powerful **I**nstance-**C**onditioned **A**daptation **M**odel (ICAM) to improve the large-scale generalization performance for RL-based NCO. Our contributions can be summarized as follows:

- We design a simple yet efficient instance-conditioned adaptation function to adaptively incorporate the geometric structure of cross-scale instances with a small computational overhead.

- We propose a powerful yet low-complexity Adaptation Attention Free Module (AAFM) to explicitly capture instance-specific features into the NCO inference process.

- We conduct a thorough experimental study to show ICAM can achieve promising generalization performance on different large-scale TSP, CVRP, and ATSP instances with a very fast inference time.

Table 1: Comparison between our ICAM and existing RL-based neural vehicle routing solvers with information incorporation.

| Neural Vehicle Routing Solvers | Information | | Module | | | Varying-scale Training |
| | Scale | Node-to-node distances | Embedding† | Attention | Compatibility | |
| --- | --- | --- | --- | --- | --- | --- |
| S2V-DQN (Khalil et al., 2017) | × | × | × | × | × | ✓ |
| DAN (Cao et al., 2021) | × | × | × | × | × | ✓ |
| SCA (Kim et al., 2022) | ✓ | × | ✓ | × | × | × |
| Meta-AM (Manchanda et al., 2022) | × | × | × | × | × | ✓ |
| Pointerformer (Jin et al., 2023) | × | ✓ | × | ✓‡ | × | × |
| Meta-SAGE (Son et al., 2023) | ✓ | ✓ | ✓ | × | ✓ | × |
| FER (Li et al., 2023a) | × | ✓ | ✓ | × | × | × |
| Omni_VRP (Zhou et al., 2023) | × | × | × | × | × | ✓ |
| ELG (Gao et al., 2024) | × | ✓ | × | × | ✓ | × |
| DAR (Wang et al., 2024) | × | ✓ | × | × | ✓ | ✓ |
| ICAM (Ours) | ✓ | ✓ | ✓ | ✓ | ✓ | ✓ |

† The embedding includes node embedding and context embedding. In FER, the information is used to refine node embeddings via an extra network, and SCA and Meta-SAGE use the scale information to update context embedding. Unlike them, ICAM updates node embeddings by incorporating information into the attention calculations in the encoding phase.
‡ In Pointerformer, node-to-node distances are used in the attention calculation of the decoder but are not employed in the encoder.

## 2 INSTANCE-CONDITIONED ADAPTATION

### 2.1 MOTIVATION AND KEY IDEA

For solving routing problems, the instance-specific pattern could be very helpful in finding a better solution for each instance. As shown in Figure 1, with different numbers of nodes, the geometric structures of two instances and their optimal solutions are quite different, which could provide valuable information for the solvers. For classic heuristic algorithms, the node-to-node distance information has been utilized to adapt the search behaviors for different instances (Yu et al., 2009; Arnold & Sörensen, 2019).

The instance-specific information has also been leveraged by different RL-based NCO methods as shown in Table 1. However, they still struggle to achieve a satisfying generalization performance, especially for large-scale instances. We provide a detailed review of different information incorporation strategies in Appendix A. By systematically analyzing the existing works, we find that the following three aspects are very important in properly incorporating the instance-conditioned information into the NCO model:

- **Effectively Leverage Instance-conditioned Information:** Given the diverse geometric structures and patterns of instances across different scales, effectively capturing the instance-specific features (e.g., distance and scale) is crucial for achieving good generalization performance.

- **Multiple Modules Integration:** Incorporating instance-conditioned information into multiple modules (e.g., embedding, attention, and compatibility) can make the model better aware of instance-specific information throughout the solution construction process.

- **Expanding Training Scale:** Training the NCO model on instances with a large scale range is very helpful in learning more scale-independent features, thereby achieving better large-scale generalization performance.

In the following subsections, we describe in detail how the proposed ICAM effectively obtains a better generalization performance on routing instances with different scales.

### 2.2 INSTANCE-CONDITIONED ADAPTATION FUNCTION

In this work, we propose a straightforward yet efficient instance-conditioned adaptation function $f(N, d_{ij})$ to incorporate the instance-specific information into the NCO model:

$$f(N, d_{ij}) = -\alpha \cdot \log_2 N \cdot d_{ij} \quad \forall i, j \in 1, \ldots, N, \tag{1}$$

where $N$ is the scale information (e.g., the total number of nodes), $d_{ij}$ represents the distance between node $i$ and node $j$, and $\alpha > 0$ is the learnable parameter. We take the logarithm for scale $N$

to avoid extremely high values on large-scale instances. According to the definition, this adaptation function should have a larger score for a nearer distance $d_{ij}$. As shown in Figure 2, by providing $f(N, d_{ij})$ in the whole neural solution construction process, the model is expected to be better aware of the instance-specific information and hence generate a better solution for each instance.

It can be seen that the proposed function imposes only one learnable parameter to enable the model to automatically learn the degree of adaptability across varying-scale instances. Compared with recent works that also incorporate auxiliary information, our function has the following advantages:

- We effectively leverage scale and node-to-node distances that are specific to the instances to incorporate the geometric structures of cross-scale instances.
- By incurring small time and memory overhead, the function enables the model to keep a high efficiency when facing large-scale instances.

**Less Is More**   To demonstrate the superiority of our proposed function $f(N, d_{ij})$, we report its performance on solving the TSP1000 instances using the seminal POMO model (Kwon et al., 2020), and compare it with three typical information incorporation approaches: (1) Simple node-to-node distances (Jin et al., 2023; Wang et al., 2024); (2) Node-to-node distances with a bias coefficient $\alpha$ introduced (Son et al., 2023); and (3) An extra local policy as adopted in Gao et al. (2024). As shown in Table 2, our proposed function can significantly improve the generalization performance of the original model with a small time and memory overhead. For detailed experimental settings and results, please refer to Appendix B.

Table 2: Comparison on TSP1000 instances with different instance-specific information incorporation approaches.

| Method | Params | Avg.memory | Gap | Time |
|---|---|---|---|---|
| POMO | 1.27M | 107.50MB | 25.916% | 63.80 s |
| POMO + dist. | 1.27M | 124.22MB | 22.696% | 83.85s |
| POMO + $\alpha$ * dist. | 1.27M | 124.22MB | 14.517% | 86.23s |
| POMO + Local policy | 1.30M | 163.44MB | 14.821% | 130.26s |
| POMO + $f(N, d_{ij})$ | 1.27M | 124.22MB | **10.812%** | 86.92s |

### 2.3   Instance-Conditioned Adaptation Model

In addition to the instance-conditioned adaptation function, the NCO model structure is also crucial to achieve a promising generalization performance. Most existing models adopt the encoder-decoder structure, which is developed from Transformer (Kool et al., 2019; Gao et al., 2024). Without loss of generality, taking well-known POMO (Kwon et al., 2020) as an example, this subsection briefly reviews the prevailing neural solution construction pipeline and discusses how to efficiently incorporate the instance-specific information.

**Rethinking Attention Mechanism in NCOs**   Given an instance $S = \{\mathbf{s}_i\}_{i=1}^N$, $\mathbf{s}_i$ represents the features of each node (e.g., the coordinates of each city in TSPs). These features are transformed into initial embeddings $H^{(0)} = (\mathbf{h}_1^{(0)}, \ldots, \mathbf{h}_N^{(0)})$ via a linear projection. The initial embeddings pass through $L$ attention layers to get node embeddings $H^{(L)} = (\mathbf{h}_1^{(L)}, \ldots, \mathbf{h}_N^{(L)})$. The attention layer consists of a Multi-Head Attention (MHA) sub-layer (Vaswani et al., 2017) and a Feed-Forward (FF) sub-layer. During the decoding process, POMO model generates a solution in an autoregressive manner. For the example of TSP, in the $t$-step construction, the context embedding is composed of the first visited node embedding and the last visited node embedding, i.e., $\mathbf{h}_{(C)}^t = [\mathbf{h}_{\pi_1}^{(L)}, \mathbf{h}_{\pi_{t-1}}^{(L)}]$. The new context embedding $\hat{\mathbf{h}}_{(C)}^t$ is then obtained via the MHA operation on $\mathbf{h}_{(C)}^t$ and $H^{(L)}$. Finally, the model yields the selection probability for each unvisited node $p_{\boldsymbol{\theta}}(\pi_t = i \mid S, \pi_{1:t-1})$ by calculating compatibility on $\hat{\mathbf{h}}_{(C)}^t$ and $H^{(L)}$.

From the above description, MHA operation is the core component of Transformer-like NCO models. In the mode of self-attention, MHA performs a scaled dot-product attention for each head. The self-attention calculation is written as

$$Q = XW^Q, \quad K = XW^K, \quad V = XW^V, \tag{2}$$

$$\text{Attention}(Q, K, V) = \text{softmax}\left(\frac{QK^{\mathrm{T}}}{\sqrt{d_k}}\right)V, \tag{3}$$

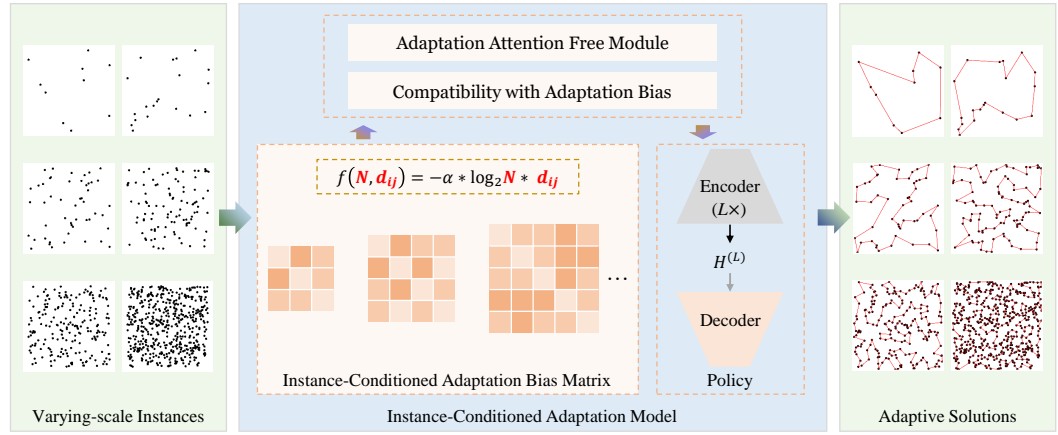

Figure 2: The proposed ICAM. Taking the TSP as an example, comprehensive instance-conditioned information is incorporated into the whole solution construction process. ICAM solves the specific instance by adaptively updating the corresponding adaptation bias matrix. Specifically, we utilize AAFM to replace all MHA operations and combine $f(N, d_{ij})$ with the compatibility calculation.

where $X$ represents the input, $W^Q$, $W^K$ and $W^V$ are three learning matrices, and $d_k$ is the dimension for $K$. In a Transformer-based NCO model, the MHA incurs primary memory usage and computational cost. In addition, the MHA calculation is not convenient for capturing the relationship between nodes. It cannot directly take advantage of the pair-wise distances between nodes.

**Adaptation Attention Free Module**   As shown in Figure 2, the proposed ICAM is also developed from the encoder-decoder structure, we remove all high-complexity MHA operations in both the encoder and decoder, and replace them with the proposed novel module, named **A**daptation **A**ttention **F**ree **M**odule (AAFM). AAFM is based on the AFT-full operation (Zhai et al., 2021), which offers more excellent speed and memory efficiency than MHA. Further details about AFT are available in Appendix C. As shown in Figure 3, the proposed AAFM can be expressed as

$$\text{AAFM}(Q, K, V, A) = \sigma(Q) \odot \frac{\exp(A)(\exp(K) \odot V)}{\exp(A)\exp(K)}, \tag{4}$$

where $Q, K, V$ are also separately obtained via Equation (2), $\sigma$ represents Sigmoid function, $\odot$ represents the element-wise product, and $A = \{\mathbf{a}_{ij}\}, \forall i, j \in 1, \ldots, N$ denotes the pair-wise adaptation bias computed by our adaptation function $f(N, d_{ij})$ in Equation (1).

Through the proposed AAFM, the model is enabled to learn instance-specific knowledge via updating pair-wise adaptation biases. Unlike traditional MHA-based NCO models, AAFM-based ICAM explicitly captures relative position biases between different nodes via adaptation function $f(N, d_{ij})$. This ability to maintain direct interaction between any two nodes in the context is a major advantage of AAFM. Furthermore, AAFM exhibits a lower computational overhead than MHA, resulting in a lower complexity and faster model.

To investigate the effectiveness of AAFM compared to MHA for information integration, we train two different models in the same settings, both adding the proposed adaptation function. The only difference between the two models is the attention mechanism (AAFM vs. MHA). For detailed analysis and experimental results, please refer to Appendix D.

**Compatibility with Adaptation Bias**   To further improve the solving performance, we integrate $f(N, d_{ij})$ into the compatibility calculation (Son et al., 2023; Gao et al., 2024). The new compatibility $u_i^t$ can be expressed as

$$u_i^t = \begin{cases} \xi \cdot \tanh(\frac{\hat{\mathbf{h}}_{(C)}^t (\mathbf{h}_i^{(L)})^{\mathrm{T}}}{\sqrt{d_k}} + a_{t-1,i}) & \text{if } i \notin \{\pi_{1:t-1}\} \\ -\infty & \text{otherwise} \end{cases}, \tag{5}$$

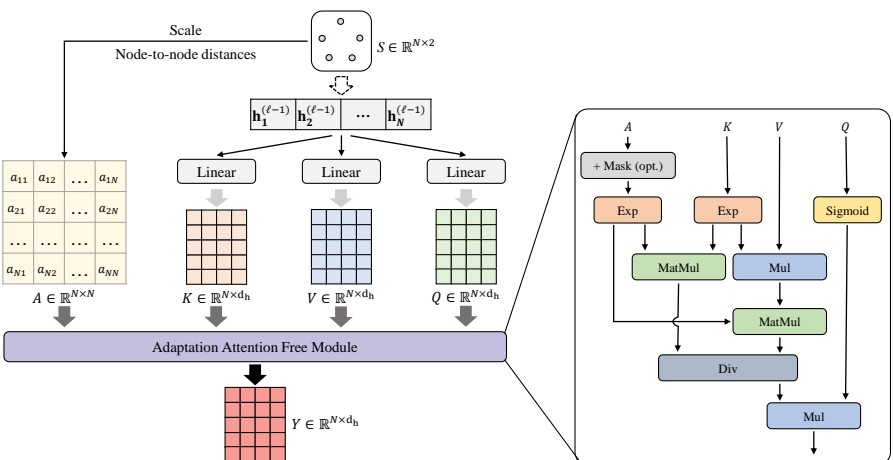

Figure 3: The proposed AAFM.

$$p_{\boldsymbol{\theta}}(\pi_t = i \mid S, \pi_{1:t-1}) = \frac{e^{u_i^t}}{\sum_{j=1}^{N} e^{u_j^t}}, \tag{6}$$

where $\xi$ is the clipping parameter, $\hat{\mathbf{h}}_{(C)}^t$ and $\mathbf{h}_i^{(L)}$ are calculated via AAFM instead of MHA. $a_{t-1,i}$ represents the adaptation bias between each remaining node and the current node.

## 3 EXPERIMENTS

In this section, we conduct a comprehensive comparison between ICAM and other classical and learning-based solvers using Traveling Salesman Problem (TSP), Capacitated Vehicle Routing Problem (CVRP), and Asymmetric Traveling Salesman Problem (ATSP) instances of different scales.

**Problem Setting**  For all problems, the instances of training and testing are generated randomly. Specifically, we generate the instances with a setup as prescribed in Kool et al. (2019) for TSPs and CVRPs, and we follow the data generation method in MatNet (Kwon et al., 2021) for ATSP. For the test set, unless stated otherwise, we generate $10,000$ instances for the 100-node case and 128 instances for cases with the scale is 200, 500, etc., the scale is up to $5,000$ for TSP and CVRP and $1,000$ for ATSP[1]. Specifically, for capacity settings in CVRP, we follow the approach in Luo et al. (2023) for scale $\leq 1,000$ and Hou et al. (2022) for scale $>1,000$, respectively.

**Model Setting**  Our proposed function $f(N, d_{ij})$ and AAFM are adaptable to different models according to the specific problem. For TSPs and CVRPs, ICAM is developed from the well-known POMO model (Kwon et al., 2020). Considering the specificity of ATSPs, we replace the backbone network with MatNet (Kwon et al., 2021). More details about the model architecture can be found in Appendix E. For all experiments, the embedding dimension is set to 128, and the dimension of the feed-forward layer is set to $512$. We set the number of attention layers in the encoder to $12$[2]. The clipping parameter $\xi = 50$ in Equation (5) for better training convergence (Jin et al., 2023). We train and test all experiments using a single NVIDIA GeForce RTX 3090 GPU with 24GB memory.

**Training**  For all models, we use Adam (Kingma & Ba, 2014) as the optimizer and the initial learning rate $\eta$ is set to $10^{-4}$. Every epoch, we process $1,000$ batches for all problems. For each instance, $N$ different solutions are always generated in parallel, following in  Kwon et al. (2020).

---

[1] For ATSP, due to memory constraints, we are unable to generate instances with scale $> 1000$ under the data generation method of MatNet, so the maximum scale for testing is $1,000$.

[2] For ATSP model, the 12-layer encoder represents two independent 6-layer encoders, following MatNet architecture (Kwon et al., 2021)

To enable the model to be aware of the scale information better and simultaneously learn various pair-wise biases of training instances at different scales, we develop a three-stage training scheme to enable the proposed ICAM to incorporate instance-conditioned information more effectively. The detailed settings of proposed three-stage training scheme are as follows:

1. **Stage 1: Warming-up on Small-scale Instances.** Initially, the model is trained for several epochs on small-scale instances. We use instances for a scale of 100 to train corresponding models for 100 epochs. Due to memory constraints, we set different batch sizes for different problems: 256 for (A)TSP and 128 for CVRP. Additionally, the capacity for CVRP instances are fixed at 50. A warm-up training can make the model more stable in the subsequent varying-scale training.

2. **Stage 2: Learning on Varying-scale Instances.** In the second stage, we train the model on varying-scale instances for much longer epochs, and for each batch, the scale $N$ is randomly sampled from the discrete uniform distribution **Unif**([100,500]) for all problems. Considering GPU memory constraints, we decrease the batch size with the scale increases. For (A)TSP, the batch size $bs = \left\lceil 160 \times (\frac{100}{N})^2 \right\rceil$. In the case of CVRP, the batch size $bs = \left\lceil 128 \times (\frac{100}{N})^2 \right\rceil$. We train the TSP model for $2,200$ epochs and CVRP model for $700$ epochs in this stage. For ATSP model, the training duration is 100 epochs attributed to the fast convergence. Furthermore, the capacity of each batch is consistently set by randomly sampling from **Unif**([50,100]) for CVRP. Under the POMO structure, $N$ trajectories are constructed in parallel for each instance during training. The loss function (denoted as $\mathcal{L}_{\text{POMO}}$) used in the first and second stages is the same as in POMO (Kwon et al., 2020).

3. **Stage 3: Top-$k$ Elite Training.** In the third stage, we want the model to focus more on the best $k$ trajectories among all $N$ trajectories. To achieve this, we design a new loss $\mathcal{L}_{\text{Top}}$, $\mathcal{L}_{\text{Top}}$ only focus on the $k$ best trajectories out of $N$ trajectories (See Equation (13)). We combine $\mathcal{L}_{\text{Top}}$ with $\mathcal{L}_{\text{POMO}}$ as the joint loss in the training of the third stage, i.e.,

$$\mathcal{L}_{\text{Joint}} = \mathcal{L}_{\text{POMO}} + \beta \cdot \mathcal{L}_{\text{Top}}. \tag{7}$$

where $\beta \in [0, 1]$ is a coefficient balancing the original loss and the new loss, $\beta$ and $k$ are set to 0.1 and 20, respectively. We adjust the learning rate $\eta$ to $10^{-5}$ across all models to enhance model convergence and training stability. The training period is standardized to 200 epochs for all models, and other settings are consistent with the second stage.

Note that for each problem, we use the same model on all scales and distributions. For more details about the model and training settings, please refer to Appendix F.

**Baseline** We compare ICAM with the following methods: (1) **Classical solver**: Concorde (Applegate et al., 2006), LKH3 (Helsgaun, 2017), HGS (Vidal, 2022) and OR-Tools (Perron & Furnon, 2023); (2) **Constructive NCO**: POMO (Kwon et al., 2020), MatNet (Kwon et al., 2021), MDAM (Xin et al., 2021), ELG (Gao et al., 2024), Pointerformer (Jin et al., 2023), Omni_VRP (Zhou et al., 2023), BQ (Drakulic et al., 2023), LEHD (Luo et al., 2023) and IN-ViT (Fang et al., 2024); (3) **Two-stage NCO**: Att-GCN+MCTS (Fu et al., 2021), DIMES (Qiu et al., 2022), TAM (Hou et al., 2022), SO (Cheng et al., 2023), DIFUSCO (Sun & Yang, 2023), H-TSP (Pan et al., 2023), T2T (Li et al., 2023b) and GLOP (Ye et al., 2024).

**Metrics and Inference** We use objective values of different solutions, optimality gaps, and total inference times to evaluate each method. Specifically, the optimality gap measures the discrepancy between the solutions generated by learning and non-learning methods and the optimal solutions, which are obtained using LKH3 for all problems. Note that times for classical solvers, which run on a single CPU, and for learning-based methods, which utilize GPUs, are inherently different. Therefore, these times should not be directly compared.

For most NCO baseline methods, we directly execute the source code provided by authors using default settings. Note that the results marked with an asterisk (*) are directly obtained from corresponding papers. For INViT, we use the INViT-3V model, and the instance augmentation is unified to aug×8, which is consistent with other methods. For TSPs and CVRPs, following Kwon et al. (2020), we report three types of results: using a single trajectory, the best result from multiple trajectories, and results derived from instance augmentation. For ATSPs, we remove instance aug-

mentation and only report the best result from multiple trajectories using a greedy strategy rather than sampled ones as adopted by MatNet.

**Results on VRPs with Scale** $\leq 1,000$    The experimental results on TSP, CVRP and ATSP with uniform distribution and scale $\leq 1,000$ are reported in Table 3. Our method stands out for consistently delivering superior inference performance, complemented by remarkably fast inference times, across various problem instances. Although it cannot surpass Att-GCN+MCTS on TSP100, POMO on CVRP100, and MatNet on ATSP100, the time it consumes is significantly less, such as Att-GCN+MCTS takes 15 minutes compared to our 37 seconds and MatNet requires over an hour compared to our 7s. On TSP1,000, our model impressively reduces the optimality gap to less than 3% in just 2 seconds. When switching to a multi-greedy strategy, the optimality gap further narrows to 1.9% in 30 seconds. With the instance augmentation, ICAM can achieve the optimality gap of 1.58% in less than 4 minutes. For a fair comparison, we have adjusted the number of RRC interaction for LEHD and the width of beam search for BQ such that all methods have a similar inference time. According to the results, ICAM can obtain a better generalization performance than LEHD and RRC on most comparisons. To the best of our knowledge, for TSP, CVRP and ATSP up to 1,000 nodes, ICAM shows state-of-the-art performance among all RL-based constructive NCO methods.

**Results on Cross-distribution VRP Instances**    We use the TSP/CVRP1,000 datasets with rotation and explosion distributions to evaluate the cross-distribution performance of ICAM. As shown in Table 4, ICAM can still achieve the best performance on specific distribution instances and the fastest speed of all comparable models. These results confirm that the same adaptation function $f(N, d_{ij})$ can perform well across problem instances with different distributions.

**Results on VRPs with Scale** $>1,000$    We also conduct experiments on instances for TSP and CVRP with larger scales, the instance augmentation is not employed for all methods due to computational efficiency. As shown in Table 5, for CVRP on all instances except for CVRP3K, ICAM outperforms all comparable methods, including INViT, GLOP with LKH3 solver and all TAM variants. ICAM is slightly worse than SL-based LEHD on CVRP3K, it consumes much more solving time than ICAM. However, the superiority of ICAM is not so obvious on TSP instances with scale $>1$K (see Appendix G). Our performance is slightly worse than the two SL-based BQ and LEHD. INViT shows remarkable performance on TSP instances with scale $>1,000$ thanks to the small search space at each construction step. Nevertheless, except for TSP5K, we achieve the second best results in RL-based constructive methods. We are slightly worse than ELG on TSP5K instances, but ELG requires a longer ($4\times$) runtime due to its heavy local policy at each construction step. Overall, our method still has a good large-scale generalization.

**Results on Benchmark Dataset**    We further evaluate the performance using well-known benchmark datasets from CVRPLIB Set-X (Uchoa et al., 2017) with scale $\leq 1000$, Set-XXL (Arnold et al., 2019) with scale $\in [3000, 7000]$, and TSPLIB (Reinelt, 1991) with scale $\leq 5000$. The results are presented in Appendix H, showing that ICAM achieves the best performance of all scale ranges in Set-X and Set-XXL. In TSPLIB datasets with scale $\leq 1000$, our method is slightly worse than SL-based models (i.e., BQ and LEHD) and ELG, which has a heavy local policy at each construction step. In TSPLIB datasets with scale $>1000$, ICAM can also obtain competitive performance. Notably, ICAM has the shortest average time on TSPLIB datasets with scale $\leq 5000$ among all models. These results also show the outstanding generalization of ICAM. To the best of our knowledge, ICAM achieves the best performance among all constructive methods in the Set-X with scale $\leq 1000$ and CVRPLIB Set-XXL (Arnold et al., 2019) with scale $\in [3000, 7000]$.

## 4    ABLATION STUDY

To demonstrate the efficiency of ICAM, we conduct a detailed ablation study, mainly including:

1. **Effects of components of adaptation function** (see Appendix I.1);
2. **Effects of adaptation function** (see Appendix I.2);
3. **Effects of different stages** (see Appendix I.3);
4. **Effects of deeper encoder** (see Appendix I.4);

Table 3: Experimental results on routing problems (TSP, CVRP, and ATSP) with uniform distribution and scale $\leq 1,000$.

| Method | TSP100 Obj. | Gap | Time | TSP200 Obj. | Gap | Time | TSP500 Obj. | Gap | Time | TSP1000 Obj. | Gap | Time |
|---|---|---|---|---|---|---|---|---|---|---|---|---|
| LKH3 | 7.7632 | 0.000% | 56m | 10.7036 | 0.000% | 4m | 16.5215 | 0.000% | 32m | 23.1199 | 0.000% | 8.2h |
| Concorde | 7.7632 | 0.000% | 34m | 10.7036 | 0.000% | 3m | 16.5215 | 0.000% | 32m | 23.1199 | 0.000% | 7.8h |
| Att-GCN+MCTS* | **7.7638** | **0.037%** | 15m | 10.8139 | 0.884% | 2m | 16.9655 | 2.537% | 6m | 23.8634 | 3.224% | 13m |
| DIMES AS+MCTS* | – | – | – | – | – | – | 16.84 | 1.76% | 2.15h | 23.69 | 2.46% | 4.62h |
| SO-mixed* | – | – | – | 10.7873 | 0.636% | 21.3m | 16.9431 | 2.401% | 32m | 23.7656 | 2.800% | 55.5m |
| DIFUSCO greedy+2-opt* | 7.78 | 0.24% | – | – | – | – | 16.80 | 1.49% | 3.65m | 23.56 | 1.90% | 12.06m |
| T2T sampling* | – | – | – | – | – | – | 17.02 | 2.84% | 15.98m | 24.72 | 6.92% | 53.92m |
| H-TSP | – | – | – | – | – | – | 17.549 | 6.220% | 23s | 24.7180 | 6.912% | 47s |
| GLOP (more revisions) | 7.7668 | 0.046% | 1.9h | 10.7735 | 0.653% | 42s | 16.8826 | 2.186% | 1.6m | 23.8403 | 3.116% | 3.3m |
| BQ greedy | 7.7903 | 0.349% | 1.8m | 10.7644 | 0.568% | 9s | 16.7165 | 1.180% | 46s | 23.6452 | 2.272% | 1.9m |
| BQ bs4 | 7.7691 | 0.076% | 4.3m | **10.7321** | **0.266%** | 21s | 16.6530 | 0.796% | 1.9m | 23.5090 | 1.683% | 4.6m |
| LEHD greedy | 7.8080 | 0.577% | 27s | 10.7956 | 0.859% | 2s | 16.7792 | 1.560% | 16s | 23.8523 | 3.168% | 1.6m |
| LEHD RRC10 | 7.7746 | 0.146% | 1.8m | 10.7431 | 0.369% | 8s | 16.6702 | 0.900% | 1.2m | 23.5894 | 2.031% | 5.5m |
| MDAM bs50 | 7.7933 | 0.388% | 21m | 10.9173 | 1.996% | 3m | 18.1843 | 10.065% | 11m | 27.8306 | 20.375% | 44m |
| POMO aug×8 | 7.7736 | 0.134% | 1m | 10.8677 | 1.534% | 5s | 20.1871 | 22.187% | 1.1m | 32.4997 | 40.570% | 8.5m |
| ELG aug×8 | 7.7807 | 0.225% | 3m | 10.8620 | 1.480% | 13s | 17.6544 | 6.857% | 2.3m | 25.5769 | 10.627% | 15.4m |
| Pointerformer aug×8 | 7.7759 | 0.163% | 49s | 10.7796 | 0.710% | 11s | 17.0854 | 3.413% | 53s | 24.7990 | 7.263% | 6.4m |
| ICAM single trajec. | 7.8328 | 0.897% | 2s | 10.8255 | 1.139% | <1s | 16.7777 | 1.551% | 1s | 23.7976 | 2.931% | 2s |
| ICAM | 7.7991 | 0.462% | 5s | 10.7753 | 0.669% | <1s | 16.6978 | 1.067% | 4s | 23.5608 | 1.907% | 28s |
| ICAM aug×8 | 7.7747 | 0.148% | 37s | 10.7385 | 0.326% | 3s | **16.6488** | **0.771%** | 38s | **23.4854** | **1.581%** | 3.8m |

| Method | CVRP100 Obj. | Gap | Time | CVRP200 Obj. | Gap | Time | CVRP500 Obj. | Gap | Time | CVRP1000 Obj. | Gap | Time |
|---|---|---|---|---|---|---|---|---|---|---|---|---|
| LKH3 | 15.6465 | 0.000% | 12h | 20.1726 | 0.000% | 2.1h | 37.2291 | 0.000% | 5.5h | 37.0904 | 0.000% | 7.1h |
| HGS | 15.5632 | -0.533% | 4.5h | 19.9455 | -1.126% | 1.4h | 36.5611 | -1.794% | 4h | 36.2884 | -2.162% | 5.3h |
| GLOP-G (LKH3) | – | – | – | – | – | – | – | – | – | 39.6507 | 6.903% | 1.7m |
| BQ greedy | 16.0730 | 2.726% | 1.8m | 20.7722 | 2.972% | 10s | 38.4383 | 3.248% | 47s | 39.2757 | 5.892% | 1.9m |
| BQ bs4 | 15.9073 | 1.667% | 4.3m | 20.4879 | 1.563% | 22s | 37.8951 | 1.789% | 1.9m | 38.5503 | 3.936% | 4.7m |
| LEHD greedy | 16.2173 | 3.648% | 30s | 20.8407 | 3.312% | 9s | 38.4125 | 3.178% | 17s | 38.9122 | 4.912% | 1.6m |
| LEHD RRC10 | 15.8892 | 1.551% | 2.2m | 20.4638 | 1.443% | 9s | 37.8564 | 1.685% | 1.5m | 38.5287 | 3.878% | 4.3m |
| MDAM bs50 | 15.9924 | 2.211% | 25m | 21.0409 | 4.304% | 3m | 41.1376 | 10.498% | 12m | 47.4068 | 27.814% | 47m |
| POMO aug×8 | **15.7544** | **0.689%** | 1.2m | 21.1542 | 4.866% | 6s | 44.6379 | 19.901% | 1.2m | 84.8978 | 128.894% | 9.8m |
| ELG aug×8 | 15.8382 | 1.225% | 4.4m | 20.6787 | 2.509% | 19s | 39.2602 | 5.456% | 3m | 41.3046 | 11.362% | 19.4m |
| ICAM single trajec. | 16.1868 | 3.453% | 2s | 20.7509 | 2.867% | <1s | 37.9594 | 1.962% | 1s | 38.9709 | 5.070% | 2s |
| ICAM | 15.9386 | 1.867% | 7s | 20.5185 | 1.715% | 1s | 37.6040 | 1.007% | 5s | 38.4170 | 3.577% | 35s |
| ICAM aug×8 | 15.8720 | 1.442% | 47s | **20.4334** | **1.293%** | 4s | **37.4858** | **0.689%** | 42s | **38.2370** | **3.091%** | 4.5m |

| Method | ATSP100 Obj. | Gap | Time | ATSP200 Obj. | Gap | Time | ATSP500 Obj. | Gap | Time | ATSP1000 Obj. | Gap | Time |
|---|---|---|---|---|---|---|---|---|---|---|---|---|
| LKH3 | 1.5777 | 0.000% | 17.4m | 1.6000 | 0.000% | 28s | 1.6108 | 0.000% | 2.3m | 1.6157 | 0.000% | 9m |
| OR-Tools | 1.8297 | 15.973% | 1.0h | 1.9209 | 20.056% | 4m | 2.0040 | 24.410% | 35.9m | 2.0419 | 26.379% | 3.1h |
| GLOP | 1.7705 | 12.220% | 23m | 1.9915 | 24.472% | 19s | 2.207 | 36.986% | 24s | 2.3263 | 43.980% | 52s |
| MatNet ×128 | **1.5838** | **0.385%** | 1.1h | 3.6894 | 130.588% | 4.3 m | – | – | – | – | – | – |
| ICAM | 1.6531 | 4.782% | 7s | **1.6886** | **5.537%** | 1s | **1.7343** | **7.664%** | 5s | **1.8580** | **14.994%** | 34s |

Table 4: Experimental results on cross-distribution generalization.

| Method | TSP1000, Rotation Obj. (Gap) | Time | TSP1000, Explosion Obj. (Gap) | Time | CVRP1000, Rotation Obj. (Gap) | Time | CVRP1000, Explosion Obj. (Gap) | Time |
|---|---|---|---|---|---|---|---|---|
| Optimal | 17.20 (0.00%) | – | 15.63 (0.00%) | – | 32.49 (0.00%) | – | 32.31 (0.00%) | – |
| POMO aug×8 | 24.58 (42.84%) | 8.5m | 22.70(45.24%) | 8.5m | 64.22 (97.64%) | 10.2m | 59.52 (84.24%) | 11.0m |
| Omni_VRP+FS* | 19.53(14.30%) | 49.9m | 17.75(13.38%) | 49.9m | 35.60 (10.26%) | 56.8m | 35.25 (10.45%) | 56.8m |
| ELG aug×8 | 19.09(10.97%) | 15.6m | 17.37 (11.16%) | 13.7m | 37.04(14.00%) | 20.1m | 36.48(12.92%) | 20.5m |
| ICAM | 18.97 (10.28%) | 28s | 17.35 (10.99%) | 28s | 34.72 (6.86%) | 36s | 34.67 (7.31%) | 36s |
| ICAM aug×8 | **18.81(9.34%)** | 3.8m | **17.17 (9.86%)** | 3.8m | **34.54 (6.28%)** | 4.6m | **34.50 (6.79%)** | 4.5m |

[†] All datasets are obtained from Omni_VRP(Zhou et al., 2023) and contain 128 instances, and the runtime marked with an asterisk (*) is proportionally adjusted (128/1000) to match the size of our test datasets.

5. **Effects of larger training scale** (See Appendix I.5);

6. **Effects of different $\alpha$ settings** (See Appendix I.6);

7. **Parameter settings in the third stage** (see Appendix I.7);

8. **ICAM vs. POMO with three-stage training scheme** (see Appendix I.8);

9. **Comparison under the same training setting** (see Appendix I.9);

10. **The performance of POMO-Adaptation** (see Appendix I.10);

11. **Complexity analysis** (see Appendix I.11).

Table 5: Comparison on CVRP with scale $>1,000$. "Avg.time" represents the average time per instance.

| Method | CVRP2000 | | CVRP3000 | | CVRP4000 | | CVRP5000 | |
|---|---|---|---|---|---|---|---|---|
| | Obj. (Gap) | Avg.time(s) | Obj. (Gap) | Avg.time(s) | Obj. (Gap) | Avg.time(s) | Obj. (Gap) | Avg.time(s) |
| LKH3* | 64.93 (0.00%) | 20.29 | 89.90 (0.00%) | 41.10 | 118.03 (0.00%) | 80.24 | 175.66 (0.00%) | 151.64 |
| TAM-AM* | 74.31 (14.45%) | 2.2 | – | – | – | – | 172.22 (-1.96%) | 11.78 |
| TAM-LKH3* | 64.78 (-0.23%) | 5.63 | – | – | – | – | 144.64 (-17.66%) | 17.19 |
| TAM-HGS* | – | – | – | – | – | – | 142.83 (-18.69%) | 30.23 |
| GLOP-G (LKH3) | 63.02 (-2.94%) | 1.34 | 88.32 (-1.76%) | 2.12 | 114.20 (-3.25%) | 3.25 | 140.35 (-20.10%) | 4.45 |
| LEHD greedy | 61.58 (-5.16%) | 5.69 | **86.96 (-3.27%)** | 18.39 | 112.64 (-4.57%) | 44.28 | 138.17 (-21.34%) | 87.12 |
| BQ greedy | 62.59 (-3.61%) | 1.83 | 88.40 (-1.67%) | 4.65 | 114.15 (-3.29%) | 11.50 | 139.84 (-20.39%) | 27.63 |
| INViT-3V greedy | 67.35(3.73%) | 25.15 | 94.63(5.26%) | 42.77 | 120.49( 2.09%) | 62.63 | 146.61(-16.54%) | 86.47 |
| ELG | 67.54(4.02%) | 11.43 | 94.42 (5.03%) | 30.21 | 120.10 (1.75%) | 66.59 | 145.31 (-17.28%) | 121.57 |
| ICAM single trajec. | 62.38 (-3.93%) | 0.04 | 89.06 (-0.93%) | 0.10 | 115.09 (-2.49%) | 0.19 | 140.25 (-20.16%) | 0.28 |
| ICAM | **61.34 (-5.53%)** | 2.20 | 87.20 (-3.00%) | 6.42 | **112.20 (-4.94%)** | 15.50 | **136.93 (-22.05%)** | 29.16 |

[†] The total number of CVRP instances for each scale is 100, following Hou et al. (2022). Except for CVRP3K/4K instances, the optimal values are from the original paper(Hou et al., 2022).

**Capturing Instance-specific Features**   Given the diverse variations in patterns and geometric structures across different scales, we argue that instance-conditioned adaptation is crucial for improving the generalization of NCOs. ICAM can capture deeper instance-specific features than existing models. This is one of the notable contributions of ICAM. For more detailed discussions, please refer to Appendix J.

**Efficient Inference Strategies for Different Models**   To further improve performance, many search-based inference strategies are developed for NCO models. For example, BQ employs beam search, while LEHD uses the Random Re-Construct (RRC). These strategies also improve the performance of ICAM, but the improvement is not as significant as BQ and LEHD. We report the key results with different search-based decoding methods in Appendix K for better discussion.

## 5   CONCLUSION, LIMITATION, AND FUTURE WORK

**Conclusion**   In this work, we have proposed a novel ICAM to improve large-scale generalization for RL-based NCO. we design a simple yet efficient instance-conditioned adaptation function to significantly improve the generalization performance of existing NCO models with a small time and memory overhead. Further, the instance-conditioned information is more effectively incorporated into the whole neural solution construction process via a powerful yet low-complexity AAFM and the new compatibility calculation. The experimental results on various TSP, CVRP and ATSP instances show that ICAM achieves promising generalization abilities compared with other representative methods.

**Limitation and Future Work**   Although ICAM demonstrates superior performance with greedy decoding, we have observed its poor applicability to other complex inference strategies (e.g., RRC and beam search). In the future, we will develop a suitable inference strategy for ICAM.

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

# A RELATED WORK

## A.1 NON-CONDITIONED NCO

Most NCO methods are trained on a fixed scale (e.g., 100 nodes), they usually perform well on the instances with the scale trained on, but their performance could drop dramatically on instances with different scales (Kwon et al., 2020; Xin et al., 2020; 2021). To mitigate the poor generalization performance, an extra search procedure is usually required to find a better solution. Some widely used search methods include beam search (Joshi et al., 2019; Choo et al., 2022), Monte Carlo tree search (MCTS) (Xing & Tu, 2020; Fu et al., 2021; Qiu et al., 2022; Sun & Yang, 2023), and active search (Bello et al., 2016; Hottung et al., 2022). However, these procedures are very time-consuming, could still perform poorly on instances with quite different scales, and might require expert-designed strategies on a specific problem (e.g., MCTS for TSP). Recently, some two-stage approaches (Kim et al., 2021; Hou et al., 2022; Li et al., 2021; Pan et al., 2023; Cheng et al., 2023; Ye et al., 2024) have been proposed. Although these methods have better generalization abilities, they usually require expert-designed solvers and ignore the dependency between two stages, which makes model design difficult, especially for non-expert users.

## A.2 VARYING-SCALE TRAINING IN NCO

Directly training the NCO model on instances with different scales is another popular way to improve its generalization performance. Expanding the training scale can bring a broader range of cross-scale data. Training using these data enables the model to learn more scale-independent features, thereby achieving better large-scale generalization performance. This straightforward approach can be traced back to Vinyals et al. (2015) and Khalil et al. (2017), which try to train the model on instances with varying scales to improve solving performance. Furthermore, Joshi et al. (2020) systematically tests the generalization performance of NCO models by training on different TSP instances with 20-50 nodes. Subsequently, a series of works have been developed to utilize the varying-scale training scheme to improve their own NCO models' generalization performance (Lisicki et al., 2020; Cao et al., 2021; Manchanda et al., 2022; Zhou et al., 2023). Similar to the varying-scale training scheme, a few SL-based NCO methods learn to construct partial solutions with various scales during training and achieve a robust generalization performance (Luo et al., 2023; Drakulic et al., 2023). Wang et al. (2024) train the NCO model on varying-scale instances to obtain a better generalization performance. Nevertheless, in real-world applications, it could be very difficult to obtain high-quality labeled solutions for SL-based model training. RL-based models also face the challenge of efficiently capturing cross-scale features from varying-scale training data, which severely hinders their generalization ability on large-scale problems.

## A.3 INFORMATION-CONDITIONED NCO

Recently, several works have indicated that incorporating auxiliary information (e.g., the distance between each pair of nodes for VRPs) can facilitate model training and improve solving performance. In Kim et al. (2022), the scale-related feature is added to the context embedding of the decoder to make the model scale-aware during the decoding phase. Jin et al. (2023), Son et al. (2023) and Wang et al. (2024) use the distance to bias the output score in the decoding phase, thereby guiding the model toward more efficient exploration. Especially, Gao et al. (2024) employ a local policy network to catch distance knowledge and integrate it into the compatibility calculation, and in Li et al. (2023a), the distance-related feature is utilized to refine node embeddings to improve the model exploration. None of them incorporate the information into the whole neural solution construction process and fail to achieve satisfactory generalization performance on large-scale instances.

# B  COMPARISON BETWEEN DIFFERENT INCORPORATION APPROACHES

To demonstrate the superiority of our function $f(N, d_{ij})$, TSP as an example, we train various models in the same training settings, the only difference between these models is the incorporation approaches with auxiliary information. Without loss of generality, in this experiment, all comparable models are developed from a well-known NCO model, that is POMO (Kwon et al., 2020). We train all models for 100 epochs, every epoch, we process $1,000$ batches, and the batch size $bs = 64$ for all models. The incorporation approaches mainly include:

- Simple node-to-node distances (Jin et al., 2023; Wang et al., 2024);
- Node-to-node distances with a bias coefficient $\alpha$ introduced (Son et al., 2023);
- An extra local policy as adopted in Gao et al. (2024);
- Our proposed adaptation function $f(N, d_{ij})$.

We incorporate the above four approaches into all attention calculations in both the encoder and decoder, respectively. Moreover, we also combine them with the compatibility calculation in the decoder (Gao et al., 2024; Wang et al., 2024). Considering the special design of MHA, the way that we integrate the four approaches with Self-Attention in MHA can be expressed as

$$\text{Attention}(Q, K, V) = \text{softmax}\left(\frac{QK^{\text{T}}}{\sqrt{d_k}} + G\right)V, \tag{8}$$

where $G = \{g_{ij}\}$, $\forall i, j \in 1, \ldots, N$ denotes the value via different incorporation approaches. Note that the clipping parameter is changed to $50$ for better training convergence (Jin et al., 2023), and the rest of model parameters are consistent with the original POMO model.

Table 6: Comparison between different incorporation approaches. "Avg.memory" represents the average memory usage per instance.

| Method | Model Params | TSP100 | | | TSP200 | | | TSP500 | | | TSP1000 | | |
|---|---|---|---|---|---|---|---|---|---|---|---|---|---|
| | | Avg.memory | Gap | Time | Avg.memory | Gap | Time | Avg.memory | Gap | Time | Avg.memory | Gap | Time |
| POMO | 1.27M | 1.47MB | 1.318% | 7.68 s | 5.11MB | 4.216% | 1.08 s | 28.09MB | 14.946% | 8.34 s | 107.50MB | 25.916% | 63.80 s |
| POMO + dist. | 1.27M | 1.77MB | 0.924% | 9.00s | 6.02MB | 3.461% | 1.21s | 32.62MB | 13.194% | 10.42s | 124.22MB | 22.696% | 83.85s |
| POMO + $\alpha$ *dist. | 1.27M | 1.77MB | 0.843% | 9.14s | 6.02MB | 2.913% | 1.25 s | 32.62MB | 9.550% | 10.75 s | 124.22MB | 14.517% | 86.23s |
| POMO + Local policy | 1.30M | 3.29MB | **0.659%** | 23.82 s | 9.61MB | 2.730% | 2.23s | 45.55MB | 9.587% | 18.80 s | 163.44MB | 14.821% | 130.26s |
| POMO + $f(N, d_{ij})$ | 1.27M | 1.77MB | 0.774% | 9.16s | 6.02MB | **2.442%** | 1.25 s | 32.62MB | **7.208%** | 11.18s | 124.22MB | **10.812%** | 86.92s |

As shown in Table 6, on TSP100 instances, POMO with our proposed function $f(N, d_{ij})$ performs slightly worse than POMO with an extra local policy as adopted in Gao et al. (2024), but it takes more than twice as long as ours. In addition, the generalization is significantly improved even with the simple addition of only a $\alpha$ parameter, and replacing the incorporation approach with our function $f(N, d_{ij})$ further improves its generalization performance. These impressive results highlight the effectiveness of our proposed function $f(N, d_{ij})$, compared with other approaches, our proposed function significantly improves the generalization of the original model with a very small time and memory overhead.

# C ATTENTION FREE TRANSFORMER

As a linear attention approximation mechanism, AFT (Zhai et al., 2021) offers more excellent speed and memory efficiency than MHA operation. AFT has multiple versions, and the basic version is called AFT-full. Given the input $X$, AFT first transforms it to obtain $Q, K, V$ by the corresponding linear projection operation, respectively. The calculation of AFT-full can be expressed as

$$Q = XW^Q, \quad K = XW^K, \quad V = XW^V, \tag{9}$$

$$Y_i = \sigma (Q_i) \odot \frac{\sum_{j=1}^N \exp (K_j + w_{i,j}) \odot V_j}{\sum_{j=1}^N \exp (K_j + w_{i,j})}, \tag{10}$$

where $W^Q, W^K, W^V$ are three learnable matrices, $\odot$ is the element-wise product, $\sigma$ denotes the nonlinear function applied to the query $Q$, default function is Sigmoid, $w \in \mathbb{R}^{N \times N}$ is the pair-wise position biases, and each $w_{i,j}$ is a scalar. In AFT, the model automatically updates pair-wise position biases $w$, which is used to quantify the importance of the relative position information. A detailed complexity analysis comparing AFT-full with other variants is provided in Table 7.

Table 7: Complexity comparison of AFT-Full and other AFT variants. Here $N, d, s$ denote the sequence length, feature dimension, and local window size.

| Model | Time | Space |
|---|---|---|
| Transformer | $\mathcal{O}(N^2 d)$ | $\mathcal{O}(N^2 + Nd)$ |
| AFT-full | $\mathcal{O}(N^2 d)$ | $\mathcal{O}(Nd)$ |
| AFT-simple | $\mathcal{O}(Nd)$ | $\mathcal{O}(Nd)$ |
| AFT-local | $\mathcal{O}(Nsd), \ s < N$ | $\mathcal{O}(Nd)$ |
| AFT-conv | $\mathcal{O}(Nsd), \ s < N$ | $\mathcal{O}(Nd)$ |

As shown in Table 7, the basic version of AFT outlined in Equation (10) is called AFT-full and is the version that we adopt. AFT includes three additional variants: AFT-local, AFT-simple and AFT-conv. Owing to the removal of the multi-head mechanism, compared to the traditional Transformer, AFT exhibits reduced memory usage and increased speed during both the training and testing. Further details are available in the related work section mentioned above.

# D AFT VS. MHA

In language modeling, the relation (e.g., semantic difference) between two tokens is difficult to represent directly by position bias $w_{i,j}$. According to Zhai et al. (2021), AFT obtains competitive performance but is still worse than the basic MHA operation.

However, taking the routing problem as an example, the relation between two nodes can be directly represented by only the distance information computed from the node coordinates, just as a traditional heuristic solver (e.g., LKH3 (Helsgaun, 2017)) can solve a specific instance by only inputting the distance-based adjacency matrix. In classic neural vehicle routing solvers using MHA, e.g., POMO(Kwon et al., 2020), the relation between two nodes is computed by mapping the node coordinates into a high-dimensional hidden space. In short, MHA cannot directly take advantage of the pair-wise distances between nodes.

Unlike traditional MHA operation, AFT can explicitly capture the relative position bias between different nodes via a pair-wise position bias matrix $w$. This ability to maintain direct interaction between any two nodes in the context is a major advantage of AFT. The explicit relative position information is valuable to achieve better solving performance. In fact, AFT can also be viewed as a specialized form of MHA, where each feature dimension is treated as an individual head.

To investigate the effectiveness of AFT compared to MHA in information integration, we train a new ICAM that replaces AAFM with the standard MHA, denoted as ICAM-MHA. ICAM-MHA is trained in exactly the same settings, including three-stage training, the adaptation function, model structure, and hyperparameters. The only difference between the two models is the attention mechanism (AAFM vs. MHA). The way that we integrate the adaptation function $f(N, d_{ij})$ with Self-Attention in MHA can be found in Equation (8).

Table 8: Comparison of the AFT and MHA on TSP instances with different scales.

| Method | TSP100 | | | TSP200 | | | TSP500 | | | TSP1000 | | |
|---|---|---|---|---|---|---|---|---|---|---|---|---|
| | Obj. | Gap | Time | Obj. | Gap | Time | Obj. | Gap | Time | Obj. | Gap | Time |
| Concorde | 7.7632 | 0.000% | 34m | 10.7036 | 0.000% | 3m | 16.5215 | 0.000% | 32m | 23.1199 | 0.000% | 7.8h |
| ICAM-MHA | 7.8061 | 0.552% | 10s | 10.7922 | 0.828% | 1s | 16.7613 | 1.452% | 11s | 23.7193 | 2.593% | 1.5m |
| ICAM | **7.7991** | **0.462%** | 5s | **10.7753** | **0.669%** | <1s | **16.6978** | **1.067%** | 4s | **23.5608** | **1.907%** | 28s |

As can be seen from the results in Table 8, ICAM-MHA also has good large-scale generalization performance, this again demonstrates the effectiveness of proposed adaptation function and three-stage training scheme. Further, we can observe replacing MHA with AAFM can further improve performance while significantly reducing running time. The advantages of ICAM over ICAM-MHA become more significant as the problem scale increases. The good scalability performance of ICAM may stem from the ability of AFT to integrate instance-conditioned information more efficiently.

# E MODEL ARCHITECTURE

**ICAM for TSPs and CVRPs** For the TSP and CVRP models, ICAM is an improvement based on POMO model (Kwon et al., 2020). We remove all the MHA calculations in POMO (including both the encoder and decoder) and replace them with our proposed AAFM. Additionally, as shown in Equation (5), in the decoding phase, we modify the compatibility calculation by adding our adaptation function $f(N, d_{ij})$ to the original calculation, following the approach in Gao et al. (2024) and Son et al. (2023), so as to improve the model performance further. Finally, we expand the number of encoder layers to 12 to generate better node embeddings. Note that since the heavy encoder is only called once for solution construction, there is no obvious time difference between the models with 12-layer and 6-layer Encoder.

**ICAM for ATSPs** For ATSP instances, the node coordinates are not available. Considering the special nature of ATSP, we use MatNet as the backbone network for ATSP model. Compared with the original MatNet proposed by Kwon et al. (2021), our improvements are mainly as follows:

1. In original MatNet, for initial embeddings, zero-vectors and one-hot vectors are used to embed nodes in A and nodes in B (or vice versa), respectively. However, since the embedding dimension is set to 256, this approach fails to enable MatNet to generalize to ATSP instances with more than 256 nodes efficiently. We change the dimension of the input feature to 50, i.e., the distance of the 50 nearest nodes to each node in row and column elements, respectively. Further, these features are transformed into different initial embeddings $H^{(0)} = (\mathbf{h}_1^{(0)}, \ldots, \mathbf{h}_N^{(0)})$ via different 128-dimension linear projections in 6-layer row encoder and 6-layer column encoder, respectively.

2. we also utilize AAFM to replace attention operations, including Mixed-score attention, which is proposed by MatNet in the encoding phase, and MHA operation in the decoding phase.

3. Moreover, we also combine our proposed adaptation function $f(N, d_{ij})$ with the compatibility calculation in the decoding phase.

For the ATSP model, the rest of the model architecture is consistent with MatNet, the details about MatNet can be found in Kwon et al. (2021).

# F HYPERPARAMETER AND TRAINING SETTINGS

**Model Hyperparameter Settings** The detailed information about the hyperparameter settings can be found in Table 9. Note that for the ATSP and CVRP models, we have implemented the gradient clipping technique to prevent the risk of exploding gradients.

**Training** The loss function (denoted as $\mathcal{L}_{\text{POMO}}$) used in the first and second stages is the same as in POMO (Kwon et al., 2020). According to Kwon et al. (2020), POMO is trained by the REINFORCE (Williams, 1992), and it uses gradient ascent with an approximation in Equation (11). The gradient ascent with an approximation of the loss function can be written as

$$\nabla_\theta \mathcal{L}_{\text{POMO}}(\theta) \approx \frac{1}{BN} \sum_{m=1}^{B} \sum_{i=1}^{N} R\left(\pi^i \mid S_m\right) - b^i(S_m) \nabla_\theta \log p_\theta\left(\pi^i \mid S_m\right), \quad (11)$$

$$b^i(S_m) = \frac{1}{N} \sum_{j=1}^{N} R\left(\pi^j \mid S_m\right) \quad \text{for all } i. \quad (12)$$

where $R\left(\pi^i \mid S_m\right)$ represents the total reward (e.g., the negative value of tour length) of instance $S_m$ given a specific solution $\pi^i$. Equation (12) is a shared baseline as adopted in Kwon et al. (2020).

In the third stage, we want the model to focus more on the best $k$ trajectories among all $N$ trajectories. To achieve this, we design a new loss $\mathcal{L}_{\text{Top}}$, and its gradient ascent can be expressed as

$$\nabla_\theta \mathcal{L}_{\text{Top}}(\theta) \approx \frac{1}{Bk} \sum_{m=1}^{B} \sum_{i=1}^{k} R\left(\pi^i \mid S_m\right) - b^i(S_m) \nabla_\theta \log p_\theta\left(\pi^i \mid S_m\right). \quad (13)$$

Table 9: Model hyperparameter settings in experiments.

| | TSP | CVRP | ATSP |
|---|---|---|---|
| Optimizer | | Adam | |
| Clipping parameter | | 50 | |
| Initial learning rate | | $10^{-4}$ | |
| Learning rate of stage 3 | | $10^{-5}$ | |
| Initial $\alpha$ value | | 1 | |
| Loss function of stage 1 & 2 | | $\mathcal{L}_{\mathrm{POMO}}$ | |
| Loss function of stage 3 | | $\mathcal{L}_{\mathrm{Joint}}$ | |
| Parameter $\beta$ of stage 3 | | 0.1 | |
| Parameter $k$ of stage 3 | | 20 | |
| The number of encoder layer | | 12 | |
| Embedding dimension | | 128 | |
| Feed forward dimension | | 512 | |
| Batches of each epoch | | $1,000$ | |
| Scale of stage 1 | | 100 | |
| Scale of stage 2 & 3 | | $[100, 500]$ | |
| Epochs of stage 1 | | 100 | |
| Epochs of stage 3 | | 200 | |
| Epochs of stage 2 | $2,200$ | 700 | 100 |
| Capacity of stage 1 | $-$ | 50 | $-$ |
| Capacity of stage 2 & 3 | $-$ | $[50, 100]$ | $-$ |
| Batch size of stage 1 | 256 | 128 | 256 |
| Batch size of stage 2 & 3 | $\left[160 \times (\frac{100}{N})^2\right]$ | $\left[128 \times (\frac{100}{N})^2\right]$ | $\left[160 \times (\frac{100}{N})^2\right]$ |
| Gradient clipping | $-$ | max_norm=5 | max_norm=5 |
| Weight decay | $-$ | $-$ | $10^{-6}$ |
| Total epochs | $2,500$ | $1,000$ | 400 |

We combine $\mathcal{L}_{\mathrm{Top}}$ with $\mathcal{L}_{\mathrm{POMO}}$ as the joint loss in the training of the third stage via Equation (7).

## G  RESULTS ON TSP INSTANCES WITH SCALE $>1,000$

As shown in Table 10, although ICAM equipped with adaptation biases demonstrates excellent performance and efficient inference speeds when solving TSP instances with no more than 1000 nodes, the influence of adaptation biases begins to gradually diminish as the problem scale expands beyond 1000 nodes. This phenomenon reveals an important research direction: to maintain and enhance the performance in solving larger-scale TSP instances, it is necessary to explore new strategies or improve existing adaptation strategy. This ensures that the model can effectively extend to larger problem spaces while maintaining its efficient solution-generation capabilities.

Table 10: Comparison on TSP instances with scale $>1,000$.

| | TSP2K | | | TSP3K | | | TSP4K | | | TSP5K | | |
|---|---|---|---|---|---|---|---|---|---|---|---|---|
| Method | Obj. | Gap | Avg.time (s) | Obj. | Gap | Avg.time (s) | Obj. | Gap | Avg.time (s) | Obj. | Gap | Avg.time (s) |
| LKH3 | 32.45 | 0.000% | 144.67 | 39.60 | 0.000% | 176.13 | 45.66 | 0.000% | 455.46 | 50.94 | 0.000% | 710.39 |
| LEHD greedy | 34.71 | 6.979% | 5.60 | 43.79 | 10.558% | 18.66 | 51.79 | 13.428% | 43.88 | 59.21 | 16.237% | 85.78 |
| BQ greedy | **34.03** | **4.859%** | 1.39 | 42.69 | 7.794% | 3.95 | 50.69 | 11.008% | 10.50 | 58.12 | 14.106% | 25.19 |
| INViT-3V greedy | 34.64 | 6.757% | 21.17 | **42.31** | **6.838%** | 36.23 | **48.84** | **6.965%** | 53.82 | **54.52** | **7.035%** | 74.77 |
| POMO | 50.89 | 56.847% | 4.70 | 65.05 | 64.252% | 14.68 | 77.33 | 69.370% | 35.12 | 88.28 | 73.308% | 64.46 |
| ELG | 37.12 | 14.408% | 8.17 | 45.88 | 15.855% | 23.78 | 53.35 | 16.834% | 54.27 | 59.90 | 17.594% | 101.94 |
| ICAM | 34.37 | 5.934% | 1.80 | 44.39 | 12.082% | 5.62 | 53.00 | 16.075% | 12.93 | 60.28 | 18.338% | 24.51 |

## H  RESULTS ON BENCHMARK DATASET

We further evaluate the performance using well-known benchmark datasets from CVRPLIB Set-X (Uchoa et al., 2017) (see Table 12), Set-XXL(Arnold et al., 2019)(see Table 14) and TSPLIB (Reinelt, 1991) (see Table 11 and Table 13). The results marked with an asterisk (*) are directly obtained from the original papers. Note that for scale >1000, instance augmentation is not employed for all methods due to computational efficiency.

Table 11: Experimental results on TSPLIB(Reinelt, 1991) with scale $\leq 1000$.

| Method | $N \leq 200$ (29 instances) | $200 < N \leq 500$ (13 instances) | $500 < N \leq 1000$ (6 instances) | Total (48 instances) | Avg.time |
|---|---|---|---|---|---|
| LEHD greedy | 1.92% | **3.10%** | **4.05%** | **2.51%** | 0.83s |
| BQ greedy | 2.15% | 4.35% | 4.54% | 3.04% | 2.24s |
| POMO aug×8 | 2.02% | 15.25% | 31.68% | 9.31% | 0.33s |
| INViT-3V aug×8 | 3.42% | 6.44% | 8.65% | 4.89% | 2.74s |
| ELG aug×8 | **1.18%** | 4.34% | 8.73% | 2.98% | 0.72s |
| ICAM | 4.65% | 5.77% | 12.61% | 5.95% | **0.17s** |
| ICAM aug×8 | 2.38% | 4.57% | 10.64% | 4.00% | 0.22s |

Table 12: Experimental results on CVRPLIB Set-X(Uchoa et al., 2017) with scale $\leq 1000$.

| Method | $N \leq 200$ (22 instances) | $200 < N \leq 500$ (46 instances) | $500 < N \leq 1000$ (32 instances) | Total (100 instances) | Avg.time |
|---|---|---|---|---|---|
| LEHD greedy | 11.35% | 9.45% | 17.74% | 12.52% | 1.58s |
| BQ greedy* | – | – | – | 9.94% | – |
| POMO aug×8 | 6.90% | 15.04% | 40.81% | 21.49% | 1.00s |
| INViT-3V aug×8 | 9.30% | 11.99% | 12.18% | 11.46% | 6.07s |
| ELG aug×8 | 4.51% | 5.52% | 7.80% | 6.03% | 2.56s |
| ICAM | 5.14% | 4.44% | 5.17% | 4.83% | 0.37s |
| ICAM aug×8 | **4.41%** | **3.92%** | **4.70%** | **4.28%** | 0.56s |

Table 13: Experimental results on TSPLIB (Reinelt, 1991) with scale $\leq 5,000$.

| Method | $1000 < N \leq 2000$ (15 instances) | $2000 < N \leq 3000$ (4 instances) | $3000 < N \leq 4000$ (2 instances) | $4000 < N \leq 5000$ (1 instances) | $1000 < N \leq 5000$ (22 instances) | Avg.time |
|---|---|---|---|---|---|---|
| LEHD | 10.54% | 10.93% | 13.49% | 19.05% | **11.27%** | 12.3s |
| BQ | **9.72%** | 11.58% | 24.15% | 20.35% | 11.85% | 8.9s |
| POMO | 62.76% | 64.12% | 106.61% | 66.64% | 67.17% | 6.5s |
| INViT | 12.38% | **9.11%** | **12.80%** | **7.32%** | 11.60% | 38.9s |
| ELG | 12.99% | 10.23% | 15.02% | 16.11% | 12.82% | 11.2s |
| ICAM | 13.28% | 9.88% | 14.03% | 16.79% | 12.89% | **2.8s** |

Table 14: Experimental results on CVRPLIB Set-XXL (Arnold et al., 2019) with scale $\in [3000, 7000]$.

| Method | Antwerp1 ($N = 6000$) | Antwerp2 ($N = 7000$) | Leuven1 ($N = 3000$) | Leuven2 ($N = 4000$) | Total $N \in [3000, 7000]$ | Avg.time |
|---|---|---|---|---|---|---|
| LEHD | 14.66% | 22.77% | 16.60% | 34.86% | 22.22% | 155.3s |
| BQ | 16.48% | 27.67% | 18.53% | 30.70% | 23.34% | 30.0s |
| POMO | 673.00% | 482.98% | 496.50% | 1036.64% | 672.28% | 101.9s |
| INViT | 15.40% | 27.75% | 13.71% | 26.08% | 20.74% | 90.9s |
| ELG | 13.31% | 25.53% | 16.45% | 23.25% | 19.63% | 163.3s |
| ICAM | **8.00%** | **21.66%** | **9.22%** | **15.09%** | **13.49%** | 39.9s |

# I ABLATION STUDY

Please note that, unless stated otherwise, the results presented in the ablation study reflect the best result from multiple trajectories. We do not employ instance augmentation in the ablation study, and the performance on TSP instances is used as the primary criterion for evaluation.

## I.1 EFFECTS OF COMPONENTS OF ADAPTATION FUNCTION

In our adaptation function, except for the fundamental scale and pair-wise distance information, we additionally impose a learnable parameter as well as instance scales. To better illustrate the effectiveness of this function, we conduct ablation experiments for the components, and the experimental results are shown in Table 15. The results show that both a learnable parameter $\alpha$ and scale $N$ can significantly improve the model performance.

Table 15: Comparison between component settings on TSP instances with different scales.

|  | TSP100 | TSP200 | TSP500 | TSP1000 |
|---|---|---|---|---|
| w/o learnable $\alpha$ | 0.546% | 1.124% | 2.785% | 5.232% |
| w/o scale | 0.512% | 0.866% | 2.036% | 4.236% |
| w/ learnable $\alpha$ + scale | **0.462%** | **0.669%** | **1.067%** | **1.907%** |

## I.2 EFFECTS OF ADAPTATION FUNCTION

Table 16: The detailed ablation study on instance-conditioned adaptation function. Here AFM denotes that AAFM removes the adaptation bias, and CAB is the compatibility with the adaptation bias.

|  | TSP100 | TSP200 | TSP500 | TSP1000 |
|---|---|---|---|---|
| AFM | 1.395% | 2.280% | 4.890% | 8.872% |
| AFM+CAB | 0.956% | 1.733% | 4.081% | 7.090% |
| AAFM | 0.514% | 0.720% | 1.135% | 2.241% |
| AAFM+CAB | **0.462%** | **0.669%** | **1.067%** | **1.907%** |

Given that we apply the adaptation function outlined in Equation (1) to both the AAFM and the subsequent compatibility calculation, we conducted three different experiments to validate the efficacy of this function. The data presented in Table 16 indicates a notable enhancement in the solving performance across various scales when instance-conditioned information is integrated into the model. This improvement emphasizes the importance of including detailed, fine-grained information in the model. It also highlights the critical role of explicit instance-conditioned information in improving the adaptability and generalization capabilities of RL-based models. In particular, the incorporation of richer instance-conditioned information allows the model to more effectively comprehend and address the challenges, especially in the context of large-scale problems.

## I.3 EFFECTS OF DIFFERENT STAGES

Our training is divided into three different stages, each contributing significantly to the overall effectiveness, the performance improvements achieved at each stage are detailed in Table 17. After the first stage, which uses only short training epochs, the model performs outstanding performance with small-scale instances but underperforms when dealing with large-scale instances. After the second stage, there is a marked improvement in the ability to solve large-scale instances. By the end of the final stage, the overall performance is further improved. Notably, in our ICAM, the capability to tackle small-scale instances is not affected despite the instance scales varying during the training.

Table 17: Comparsion between different stages on TSP instances with different scales.

| | TSP100 | TSP200 | TSP500 | TSP1000 |
|---|---|---|---|---|
| After stage 1 | 0.514% | 1.856% | 7.732% | 12.637% |
| After stage 2 | 0.662% | 0.993% | 1.515% | 2.716% |
| After stage 3 | **0.462%** | **0.669%** | **1.067%** | **1.907%** |

## I.4 EFFECTS OF DEEPER ENCODER

Table 18: The ablation study of encoder layers on TSP instances with different scales. Note that "L" represents encoder layers, e.g., "ICAM-6L" denotes the ICAM model using a 6-layer encoder.

| Method | Model Params | TSP100 | | TSP200 | | TSP500 | | TSP1000 | |
|---|---|---|---|---|---|---|---|---|---|
| | | Gap | Time | Gap | Time | Gap | Time | Gap | Time |
| POMO-6L | 1.27M | **0.365%** | 8s | 2.274% | 1s | 24.053% | 9s | 42.114% | 1.1m |
| ICAM-6L | **1.15M** | 0.442% | 5s | 0.722% | <1s | 1.328% | 4s | 2.422% | 28s |
| ICAM-12L | 2.24M | 0.462% | 5s | **0.669%** | <1s | **1.067%** | 4s | **1.907%** | 28s |

**The Performance with Deeper Encoder:** We have conducted an ablation study of ICAM with 6 and 12 layers, respectively. From these results, we can see that a deeper encoder structure helps the model perform better in larger-scale instances. The ICAM-6L can already significantly outperform the POMO in larger-scale TSP instances with fewer parameters. Furthermore, ICAM-12L can outperform ICAM-6L in large-scale instances.

**The Time with Deeper Encoder:** Due to our 12-layer encoder, we have more parameters than ICAM-6L. However, since the heavy encoder is only called once for the solution construction process, there is no obvious time difference between the models with 12-layer and 6-layer Encoder. Our ICAM method achieves a lower inference time for all TSPs than the POMO model.

## I.5 EFFECTS OF LARGER TRAINING SCALE

Table 19: Comparison between different training scales on TSP instances with different scales.

| Training Scale $N$ | TSP100 | TSP200 | TSP500 | TSP1000 |
|---|---|---|---|---|
| $N \in$ **Unif**($[100, 200]$) | **0.241%** | **0.461%** | 1.538% | 7.053% |
| $N \in$ **Unif**($[100, 500]$) | 0.462% | 0.669% | **1.067%** | **1.907%** |

| Training Scale $N$ | CVRP100 | CVRP200 | CVRP500 | CVRP1000 |
|---|---|---|---|---|
| $N \in$ **Unif**($[100, 200]$) | **1.542%** | **1.405%** | 1.558% | 6.300% |
| $N \in$ **Unif**($[100, 500]$) | 1.867% | 1.715% | **1.007%** | **3.577%** |

To investigate the effectiveness of training scales, we train a new model in a smaller training scale, in which the training scale $N$ is randomly sampled from **Unif**($[100,200]$). The comparison results are provided in Table 19, we can find that when we train a model on larger-scale instances, the model can obtain better performance in solving larger-scale instances. By training on larger instances, the model can see richer geometric structures and thus learn decision-making patterns for different instances, the scale diversity allows the model to perform well when facing larger-scale instances.

Similar to the experiment on TSP, we compare our proposed model with two different training scales (**Unif**($[100,200]$) or **Unif**($[100,500]$)). According to the results shown in Table 19, we can find that when we train a CVRP model on larger-scale instances, the CVRP model can also perform better in solving larger-scale instances. This observation is consistent with that for the TSP model.

## I.6 EFFECTS OF DIFFERENT $\alpha$ SETTINGS

Table 20: Comparison under different $\alpha$ settings on TSP instances with different scales. Note that all models are trained 500 epochs (i.e., 400 epochs of stage 2).

|  | TSP100 | TSP200 | TSP500 | TSP1000 |
|---|---|---|---|---|
| w/ $\alpha = 0.1$ | 1.558% | 2.841% | 6.246% | 10.304% |
| w/ $\alpha = 0.5$ | 1.077% | 2.216% | 4.797% | 8.023% |
| w/ $\alpha = 1$ | 0.843% | 1.729% | 3.898% | 6.513% |
| w/ $\alpha = 2$ | **0.820%** | 1.553% | 3.229% | 5.979% |
| w/ $\alpha = 5$ | 1.024% | 1.572% | 2.840% | 5.046% |
| w/ learnable $\alpha$ | 0.845% | **1.397%** | **2.381%** | **4.371%** |

To demonstrate the impact of the learnable parameter, we have conducted an ablation study on the value of the parameter $\alpha$. Since fixed $\alpha > 5$ will cause the exploding gradients, we keep the $\alpha$ value at a maximum of 5. Due to the time limit, all models are trained with 500 epochs and the results are shown in Table 20, the model with a learned parameter $\alpha$ can significantly outperform its counterparts with different fixed parameters.

## I.7 PARAMETER SETTINGS IN STAGE 3

In the third stage, we manually adjust the $\beta$ and $k$ values as specified in Equation (13). The experimental results for two settings involving different values are presented in Table 21. When trained using $\mathcal{L}_{\text{Joint}}$ as outlined in Equation (7), our model shows further improved performance. We observe no significant performance variation among different models at various $k$ values when using the multi-greedy search strategy. However, increasing the $\beta$ coefficients while yielding a marginal improvement in performance with the multi-greedy strategy notably diminishes the solving efficiency in the single-trajectory mode. Given the challenges in generating $N$ trajectories for a single instance as the instance scale increases, we are focusing on optimizing the model effectiveness, specifically in the single trajectory mode, to obtain the best possible performance. To avoid harming the performance under the single trajectory, we set $k$ and $\beta$ to 20 and 0.1, respectively.

Table 21: Comparsion between different parameters in the third stage on TSP1000 instances.

|  | single trajectory | | | | multiple trajectory | | | |
|---|---|---|---|---|---|---|---|---|
|  | $\beta = 0$ | $\beta = 0.1$ | $\beta = 0.5$ | $\beta = 0.9$ | $\beta = 0$ | $\beta = 0.1$ | $\beta = 0.5$ | $\beta = 0.9$ |
| $k = 20$ | 2.996% | **2.931%** | 3.423% | 3.480% | 2.039% | 1.907% | 1.859% | 1.875% |
| $k = 50$ | – | 3.060% | 3.123% | 3.328% | – | 1.935% | 1.892% | **1.857%** |
| $k = 100$ | – | 2.979% | 3.201% | 3.343% | – | 1.948% | 1.899% | 1.899% |

## I.8 ICAM VS. POMO WITH THREE-STAGE TRAINING SCHEME

To improve the ability to be aware of scale, we implement a varying-scale training scheme. Given that most of our problem models are an advancement over the POMO framework, we ensure a fair comparison by training a new POMO model using our three-stage training settings (i.e., trained on 100 to 500 nodes).

The comparison of POMO and our ICAM is provided in Table 22 to investigate the effectiveness of the proposed adaptation function. In our three-stage training scheme, POMO also obtains better generalization compared to the original model, but it is still outperformed by ICAM. According to the results, after 2500 epochs, the POMO model can obtain an optimality gap of 6.6% in TSP1000 instances. However, ICAM only requires 110 epochs to obtain a similar performance (i.e., only 10 epochs of varying-scale training) and achieve a gap of less than 2% after a complete training process. It is well known that during the training process, the later the training period, the slower the model performance improves. Therefore, this performance gain is significant but not merely a

marginal improvement. In contrast to POMO, ICAM excels in capturing cross-scale features and perceiving instance-conditioned information, this ability notably enhances model performance in solving problems across various scales.

Table 22: Comparison of ICAM and POMO with the same training settings on TSP and CVRP instances with different scales.

| Method | TSP100 | | | TSP200 | | | TSP500 | | | TSP1000 | | |
| | Obj. | Gap | Time | Obj. | Gap | Time | Obj. | Gap | Time | Obj. | Gap | Time |
|---|---|---|---|---|---|---|---|---|---|---|---|---|
| Concorde | 7.7632 | 0.000% | 34m | 10.7036 | 0.000% | 3m | 16.5215 | 0.000% | 32m | 23.1199 | 0.000% | 7.8h |
| POMO | **7.7915** | **0.365%** | 8s | 10.9470 | 2.274% | 1s | 20.4955 | 24.053% | 9s | 32.8566 | 42.114% | 1.1m |
| POMO-ThreeStage | 7.8957 | 1.707% | 8s | 10.9085 | 1.914% | 1s | 17.0488 | 3.192% | 9s | 24.6453 | 6.598% | 1.1m |
| ICAM | 7.7991 | 0.462% | 5s | **10.7753** | **0.669%** | <1s | **16.6978** | **1.067%** | 4s | **23.5608** | **1.907%** | 28s |

| Method | CVRP100 | | | CVRP200 | | | CVRP500 | | | CVRP1000 | | |
| | Obj. | Gap | Time | Obj. | Gap | Time | Obj. | Gap | Time | Obj. | Gap | Time |
|---|---|---|---|---|---|---|---|---|---|---|---|---|
| LKH3 | 15.6465 | 0.000% | 12h | 20.1726 | 0.000% | 2.1h | 37.2291 | 0.000% | 5.5h | 37.0904 | 0.000% | 7.1h |
| POMO | **15.8368** | **1.217%** | 10s | 21.3529 | 5.851% | 1s | 48.2247 | 29.535% | 10s | 143.1178 | 285.862% | 1.2m |
| POMO-ThreeStage | 16.0199 | 2.386% | 10s | 20.6401 | 2.318% | 1s | 37.8624 | 1.701% | 10s | 38.9679 | 5.062% | 1.2m |
| ICAM | 15.9386 | 1.867% | 7s | **20.5185** | **1.715%** | 1s | **37.6040** | **1.007%** | 5s | **38.4170** | **3.577%** | 35s |

## I.9    COMPARISON UNDER THE SAME TRAINING SETTING

We have now conducted the same varying-scale training with 200 epochs (VST200) for both our proposed ICAM as well as the representative RL-based POMO and ELG baselines. The SL-based LEHD and BQ are not included in this experiment since it is difficult to obtain high-quality solutions for a large amount of instances up to 500 nodes.

Table 23: Experimental results on TSPs and CVRPs with uniform distribution and scale $\leq 1,000$. Here, VST$n$ denotes this model is trained for $n$ epochs on varying-scale instances.

| Method | TSP100 | | | TSP200 | | | TSP500 | | | TSP1000 | | |
| | Obj. | Gap | Time | Obj. | Gap | Time | Obj. | Gap | Time | Obj. | Gap | Time |
|---|---|---|---|---|---|---|---|---|---|---|---|---|
| LKH3 | 7.7632 | 0.000% | 56m | 10.7036 | 0.000% | 4m | 16.5215 | 0.000% | 32m | 23.1199 | 0.000% | 8.2h |
| POMO-Original | **7.7915** | **0.365%** | 8s | 10.9470 | 2.274% | 1s | 20.4955 | 24.053% | 9s | 32.8566 | 42.114% | 1.1m |
| POMO-VST200 | 7.9820 | 2.818% | 8s | 11.0624 | 3.352% | 1s | 17.5485 | 6.216% | 9s | 25.8064 | 11.620% | 1.1m |
| ELG-Original | 7.8128 | 0.638% | 22s | 10.9512 | 2.313% | 2s | 17.8223 | 7.874% | 17s | 25.7991 | 11.588% | 2m |
| ELG-VST200 | 7.8429 | 1.027% | 22s | 10.8920 | 1.760% | 2s | 17.1632 | 3.884% | 17s | 24.7273 | 6.953% | 2m |
| ICAM-VST20 | 7.8394 | 0.982% | 5s | 10.8859 | 1.703% | <1s | 17.1075 | 3.547% | 4s | 24.6161 | 6.472% | 28s |
| ICAM-VST200 | 7.8284 | 0.840% | 5s | **10.8492** | **1.360%** | <1s | **16.9311** | **2.479%** | 4s | **24.1331** | **4.382%** | 28s |

| Method | CVRP100 | | | CVRP200 | | | CVRP500 | | | CVRP1000 | | |
| | Obj. | Gap | Time | Obj. | Gap | Time | Obj. | Gap | Time | Obj. | Gap | Time |
|---|---|---|---|---|---|---|---|---|---|---|---|---|
| LKH3 | 15.6465 | 0.000% | 12h | 20.1726 | 0.000% | 2.1h | 37.2291 | 0.000% | 5.5h | 37.0904 | 0.000% | 7.1h |
| POMO-Original | **15.8368** | **1.217%** | 10s | 21.3529 | 5.851% | 1s | 48.2247 | 29.535% | 10s | 143.1178 | 285.862% | 1.2m |
| POMO-VST200 | 16.1019 | 2.911% | 10s | 20.8046 | 3.133% | 1s | 38.3320 | 2.962% | 10s | 40.1454 | 8.237% | 1.2m |
| ELG-Original | 15.9855 | 2.166% | 34s | 20.8618 | 3.417% | 3s | 39.6746 | 6.569% | 23s | 42.0760 | 13.442% | 2.4m |
| ELG-VST200 | 16.1121 | 2.975% | 34s | 20.8045 | 3.132% | 3s | 38.3940 | 3.129% | 23s | 39.7601 | 7.198% | 2.4m |
| ICAM-VST20 | 16.0496 | 2.576% | 7s | 20.7434 | 2.830% | 1s | 38.1647 | 2.513% | 5s | 39.3221 | 6.017% | 35s |
| ICAM-VST200 | 16.0240 | 2.413% | 7s | **20.6464** | **2.349%** | 1s | **37.9161** | **1.845%** | 5s | **39.0220** | **5.208%** | 35s |

As shown in Table 23, our proposed varying-scale training (VST) method can also significantly improve the generalization performance of POMO and ELG. For example, ELG-VST200 can obtain a $6.9\%$ optimality gap on TSP1000 while the gap is $11.588\%$ for the original ELG. However, it should be emphasized that our proposed ICAM can achieve a better generalization after only 20 epochs of varying-scale training. Given the substantial variations in patterns and geometric structures across different-scale routing instances, we argue this stems from a better instance-conditioned adaptation of ICAM. These experimental results and analyses have been added in Appendix J.

## I.10    THE PERFORMANCE OF POMO-ADAPTATION

We conduct an ablation study on the three-stage training for POMO equipped with our proposed adaption function. According to Table 24, the adaption function and three-stage training scheme can significantly improve the generalization performance of POMO on large-scale problem instances. However, ICAM still performs better than POMO-Adaptation, both in terms of inference time and solution lengths.

Table 24: Experimental results of POMO using the three-stage training scheme and the adaptation function on TSP instances.

| Method | TSP100 | | | TSP200 | | | TSP500 | | | TSP1000 | | |
|---|---|---|---|---|---|---|---|---|---|---|---|---|
| | Obj. | Gap | Time | Obj. | Gap | Time | Obj. | Gap | Time | Obj. | Gap | Time |
| Concorde | 7.7632 | 0.000% | 34m | 10.7036 | 0.000% | 3m | 16.5215 | 0.000% | 32m | 23.1199 | 0.000% | 7.8h |
| POMO-Original | **7.7915** | **0.365%** | 8s | 10.9470 | 2.274% | 1s | 20.4955 | 24.053% | 9s | 32.8566 | 42.114% | 1.1m |
| POMO-Adaptation (Stage1) | 7.9803 | 2.796% | 9s | 11.1303 | 3.986% | 1s | 18.3123 | 10.839% | 11s | 26.9251 | 16.459% | 1.4m |
| POMO-Adaptation (Stage1,2) | 8.0135 | 3.224% | 9s | 11.0151 | 2.910% | 1s | 17.1872 | 4.030% | 11s | 24.6219 | 6.496% | 1.4m |
| POMO-Adaptation (Stage1,2,3) | 7.9906 | 2.929% | 9s | 10.9634 | 2.428% | 1s | 17.0508 | 3.204% | 11s | 24.2849 | 5.039% | 1.4m |
| ICAM (Stage1,2,3) | 7.7991 | 0.462% | 5s | **10.7753** | **0.669%** | <1s | **16.6978** | **1.067%** | 4s | **23.5608** | **1.907%** | 28s |

## I.11 COMPLEXITY ANALYSIS

As shown in Table 25, we report the model size, memory usage per instance, and total inference time for different RL-based constructive models. We report the complexity of the model under adopting the multi-greedy strategy. For GPU memory, we report the average GPU memory usage per instance of each method for each problem. Due to our 12-layer encoder, we have more parameters than POMO and ELG. However, since the heavy encoder is only called once for solution construction, our ICAM method achieves the lowest memory usage and the fastest inference time for all TSPs.

Table 25: Comparison between ICAM and existing works in model details. "Avg.memory" represents the average memory usage per instance. $N$ and $k$ denote the scale and the number of local neighbors, respectively.

| Method | Model Params | Time complexity | Space complexity | TSP100 | | TSP200 | | TSP500 | | TSP1000 | |
|---|---|---|---|---|---|---|---|---|---|---|---|
| | | | | Avg.memory | Time | Avg.memory | Time | Avg.memory | Time | Avg.memory | Time |
| POMO | **1.27M** | $\mathcal{O}(N^3)$ | $\mathcal{O}(N^2)$ | 1.62MB | 8s | 5.40MB | 1s | 28.82MB | 9s | 108.97MB | 1.1m |
| ELG | 1.27M | $\mathcal{O}(N^3 + N^2k)$ | $\mathcal{O}(N^2 + Nk)$ | 2.63MB | 22s | 6.29MB | 2s | 32.84MB | 17s | 126.57MB | 2m |
| ICAM | 2.24M | $\mathcal{O}(N^3)$ | $\mathcal{O}(N^2)$ | **0.89MB** | **5s** | **2.61MB** | **<1s** | **13.52MB** | **4s** | **51.69MB** | **28s** |

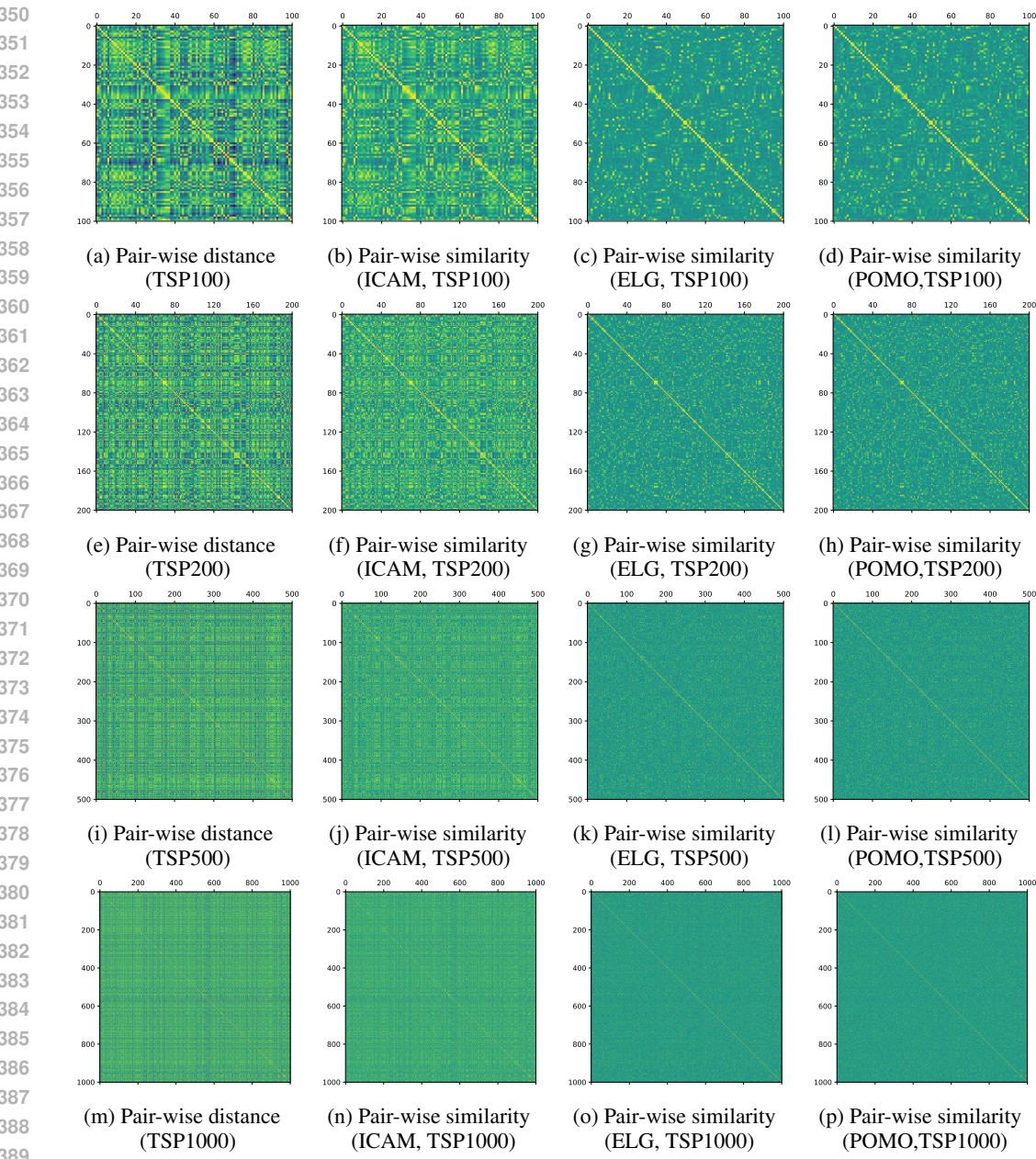

Figure 4: Comparison of cosine similarity between node embeddings generated by the encoders of different models and actual pair-wise distance with different scales. It is noteworthy that darker shades indicate lower similarity. If the node embeddings can successfully capture the instance-specific features, its similarity matrix should share some similar patterns with the normalized inverse distance matrix.

## J CAPTURING INSTANCE-SPECIFIC FEATURES

While various approaches have been explored for integrating auxiliary information, current RL-based NCO methods still struggle to achieve a satisfying generalization performance, especially for large-scale instances. The RL-based models generally adopt a heavy encoder and light decoder structure, where the quality of node embeddings generated by the encoder plays a pivotal role in overall performance. Given the diverse geometric structures and patterns of instances across dif-

ferent scales, we argue that the ability of node embeddings to adaptively capture instance-specific features across varying-scale instances is critical to improving the generalization performance.

To check whether the node embeddings can successfully capture the instance-specific features, we calculate the correlation between pair-wise node features by using the cosine similarity between node embeddings generated by the encoder. The cosine similarity calculation is defined as:

$$\text{Similarity}(e_i, e_j) = \frac{e_i \cdot e_j}{\max(\|e_i\|_2 \cdot \|e_j\|_2, \epsilon)} = \frac{\sum_{k=1}^{dim} e_{i,k} \times e_{j,k}}{\max(\sqrt{\sum_{k=1}^{dim} e_{i,k}^2} \times \sqrt{\sum_{k=1}^{dim} e_{j,k}^2}, \epsilon)} \quad (14)$$

where $e_i$ and $e_j$ represent the embeddings generated by the encoder of node $i$ and node $j$, respectively, $dim$ is the embedding dimension, $\epsilon$ is a small value to avoid division by zero ($\epsilon = 1e - 8$ in this work). It is easy to check the range of $\text{Similarity}(e_i, e_j)$ is $[-1, 1]$. A similarity value $1$ means the two compared embeddings are exactly the same, a value $-1$ means they are in the opposite direction. Once we have this similarity matrix for embeddings, we can compare it with the distance matrix of nodes to check whether they share similar patterns. For an easy visualization comparison, we can calculate the inverse distance matrix with the component $\hat{d}_{ij} = \max_{i,j} d_{ij} - d_{ij}$ and further normalize the whole matrix to the range $[-1, 1]$ via $\hat{d}_{ij} = 2 \cdot \frac{\hat{d}_{ij}}{\max_{i,j} \hat{d}_{ij}} - 1$, where a value $\hat{d}_{ij} = 1$ means node $i$ and node $i$ are at exactly the same location, and $\hat{d}_{ij} = -1$ means they are far away from each other. In this way, if the node embeddings can successfully capture the instance-specific features, its similarity matrix should share some similar patterns with the normalized inverse distance matrix.

We have conducted a case study on TSP to demonstrate the instance-conditioned adaptation ability for different models, where the results are shown in Figure 4. According to the results, the representative RL-based models (i.e., ELG and POMO) all fail to effectively capture instance-specific features in their node embeddings. On the other hand, our proposed ICAM can generate instance-conditioned node embeddings, of which the embedding correlation matrix shares similar patterns with the original distance matrix. These results clearly show that ICAM can successfully capture instance-specific features in its embeddings, which leads to its promising generalization performance.

## K  COMPARSION OF DIFFERENT INFERENCE STRATEGIES

Table 26: Experimental results with different inference strategies on TSP instances.

| Method | TSP100 | | | TSP200 | | | TSP500 | | | TSP1000 | | |
|---|---|---|---|---|---|---|---|---|---|---|---|---|
| | Obj. | Gap | Time | Obj. | Gap | Time | Obj. | Gap | Time | Obj. | Gap | Time |
| Concorde | 7.7632 | 0.000% | 34m | 10.7036 | 0.000% | 3m | 16.5215 | 0.000% | 32m | 23.1199 | 0.000% | 7.8h |
| BQ greedy | 7.7903 | 0.349% | 1.8m | 10.7644 | 0.568% | 9s | 16.7165 | 1.180% | 46s | 23.6452 | 2.272% | 1.9m |
| BQ bs16 | 7.7644 | 0.016% | 27.5m | 10.7175 | 0.130% | 2m | 16.6171 | 0.579% | 11.9m | 23.4323 | 1.351% | 29.4m |
| LEHD greedy | 7.8080 | 0.577% | 27s | 10.7956 | 0.859% | 2s | 16.7792 | 1.560% | 16s | 23.8523 | 3.168% | 1.6m |
| LEHD RRC100 | **7.7640** | **0.010%** | 16m | **10.7096** | **0.056%** | 1.2m | **16.5784** | **0.344%** | 8.7m | **23.3971** | **1.199%** | 48.6m |
| ICAM | 7.7991 | 0.462% | 5s | 10.7753 | 0.669% | <1s | 16.6978 | 1.067% | 4s | 23.5608 | 1.907% | 28s |
| ICAM RRC100 | 7.7950 | 0.409% | 2.4m | 10.7696 | 0.616% | 14s | 16.6886 | 1.012% | 2.4m | 23.5488 | 1.855% | 16.8m |
| ICAM bs16 | 7.7915 | 0.365% | 1.3m | 10.7672 | 0.594% | 14s | 16.6889 | 1.013% | 1.5m | 23.5436 | 1.833% | 10.5m |

As detailed in Table 26, we can see that upon attempting to replace the instance augmentation strategy with beam search or RRC strategies, it is observed that there is no significant improvement in the performance of our model. However, incorporating RRC technology into the LEHD model and implementing beam search technology into the BQ model both result in substantial enhancements to the performance of respective models.

We think that different model structures could require different structure-specific search-based decoding methods for efficient inference. For example, LEHD is a heavy decoder model that learns to construct partial solutions in a supervised learning manner. Therefore, the search method based on random partial solution reconstruction (RRC) could work pretty well with LEHD. On the other hand, BQ uses the bisimulation quotienting approach to reduce the state space of the MDP formulation for the combinatorial optimization problem, which exploits the symmetries of each problem for efficient problem-solving. The beam search approach can further leverage the reduced state space learned by BQ, and hence lead to promising search performance. Our proposed ICAM model leverages instance-conditioned information for efficient solution construction. However, RRC and beam search do not consider this information, which leads to a relatively smaller improvement. The design of an efficient search-based decoding method for ICAM is an important future work.

## L  LICENSES FOR USED RESOURCES

Table 27: List of licenses for the codes and datasets we used in this work

| Resource | Type | Link | License |
|---|---|---|---|
| Concorde (Applegate et al., 2006) | Code | https://github.com/jvkersch/pyconcorde | BSD 3-Clause License |
| LKH3 (Helsgaun, 2017) | Code | http://webhotel4.ruc.dk/~keld/research/LKH-3/ | Available for academic research use |
| HGS (Vidal, 2022) | Code | https://github.com/chkwon/PyHygese | MIT License |
| OR-Tools (Perron & Furnon, 2023) | Code | https://github.com/google/or-tools | Apache-2.0 License |
| H-TSP (Pan et al., 2023) | Code | https://github.com/Learning4Optimization-HUST/H-TSP | Available for academic research use |
| GLOP (Ye et al., 2024) | Code | https://github.com/henry-yeh/GLOP | MIT License |
| POMO (Kwon et al., 2020) | Code | https://github.com/yd-kwon/POMO/tree/master/NEW_py_ver | MIT License |
| ELG (Gao et al., 2024) | Code | https://github.com/gaocrr/ELG | MIT License |
| Pointerformer (Jin et al., 2023) | Code | https://github.com/pointerformer/pointerformer | Available for academic research use |
| MDAM (Xin et al., 2021) | Code | https://github.com/liangxinedu/MDAM | MIT License |
| Omni-VRP (Zhou et al., 2023) | Code | https://github.com/RoyalSkye/Omni-VRP | MIT License |
| INViT (Fang et al., 2024) | Code | https://github.com/Kasumigaoka-Utaha/INViT | Available for academic research use |
| LEHD (Luo et al., 2023) | Code | https://github.com/CIAM-Group/NCO_code/tree/main/single_objective/LEHD | Available for any non-commercial use |
| BQ (Drakulic et al., 2023) | Code | https://github.com/naver/bq-nco | CC BY-NC-SA 4.0 license |
| Cross-distribution TSPs(Zhou et al., 2023) | Dataset | https://github.com/RoyalSkye/Omni-VRP/tree/main/data/TSP/Size_Distribution | MIT License |
| Cross-distribution CVRPs(Zhou et al., 2023) | Dataset | https://github.com/RoyalSkye/Omni-VRP/tree/main/data/CVRP/Size_Distribution | MIT License |
| TSPLIB (Reinelt, 1991) | Dataset | http://comopt.ifi.uni-heidelberg.de/software/TSPLIB95/ | Available for any non-commercial use |
| CVRPLIB (Uchoa et al., 2017) | Dataset | http://vrp.galgos.inf.puc-rio.br/index.php/en/ | Available for academic research use |

We list the used existing codes and datasets in Table 27, and all of them are open-sourced resources for academic usage.

