# OpenReview forum: "ICAM: Rethinking Instance-Conditioned Adaptation in Neural Vehicle Routing Solver"
_ICLR.cc/2025/Conference — Submitted to ICLR 2025_

### Official Review · Reviewer_pvDd · 2024-10-28

**Soundness:** 3
**Presentation:** 3
**Contribution:** 3
**Rating:** 6
**Confidence:** 4

**Summary:**

This paper proposes a new method, ICAM, which aims to enhance the generalization capabilities of neural solvers on large-scale vehicle routing problem instances. Specifically, it designs an instance-conditioned adaptation function that considers both the node-to-node distances and instance scales simultaneously, and integrates it into the proposed ICAM to enrich it with auxiliary information. Furthermore, it adapts the Attention Free Transformer (AFT) and proposes the Adaptation Attention Free Module (AAFM), aiming to achieve better efficiency in handling large-scale instances. Experimental results in this paper show that the proposed ICAM can outperform prevailing competing methods.

**Strengths:**

The paper is well-written and easy to follow. Compared with previous methods, the proposed method ICAM demonstrates better optimality gap and efficiency on TSP and CVRP.

**Weaknesses:**

1. As previous works have already demonstrated that integrating node-to-node distances and instance scales into neural solvers can improve generalization performance on large-scale instances, the novelty of the proposed instance-conditioned adaptation function is very limited. Meanwhile, the proposed AAFM is also a simple adaptation from previous method AFT. Given these considerations, I am concerned that this paper may not provide valuable insights to the NCO community.
2. The three-stage training scheme seems to be crucial for achieving the experimental results reported in this paper. However, it is notable that the results from competing methods are typically obtained under much simpler settings. Although the ablation studies in Table 19 demonstrate that ICAM can outperform POMO-ThreeStage, it is also evident that three-stage training significantly enhances POMO's performance on large-scale instances. This raises a concern regarding whether other competing methods might similarly benefit from a three-stage training approach. A direct comparison using consistent training settings can make the reported experimental results much more convincing.

**Questions:**

The paper is clear and I have no questions.

---

### Official Review · Reviewer_Y1Wi · 2024-10-31

**Soundness:** 3
**Presentation:** 4
**Contribution:** 4
**Rating:** 8
**Confidence:** 4

**Summary:**

This paper proposes an instance-conditioned adaptation method for large-scale generalization of RL-based neural routing solvers. ICAM introduces a simple yet efficient adaptation function to improve the generalization of existing NCO models with a very small time and memory overhead. Additionally, a lightweight Adaptation Attention Free Module (AAFM) is integrated into the encoder-decoder structure for better incorporation of instance-conditioned information, replacing the Multi-Head Attention (MHA). Extensive experimental results show that the proposed ICAM achieves promising performance on large-scale TSP, CVRP, and ATSP instances, offering a fast inference time and significant improvement over existing methods.

**Strengths:**

1. This paper is well-written and easy to follow.

2. The proposed ICAM is a novel approach  for improving the generalization of RL-based neural routing solvers across different scales. This is an important advancement in tackling the challenges faced by current methods when dealing with large-scale instances.

3. The proposed adaptation function and model structure are both well-motivated and very useful for solving large-scale routing problems.

4. The experiments are comprehensive, demonstrating its good generalization properties.

**Weaknesses:**

1. According to Table 2, the proposed adaptation function can effectively improve the generalization of POMO with a small time and memory overhead.  Obviously, just depending on node-to-node distances is not enough, but I am curious which has a greater effect, the scale information or the learnable parameter? It would be interesting to compare the effects if either the scale information or the learnable parameter is removed.

2. This paper uses the proposed adaptation function for the proposed AAFM and subsequent compatibility calculation. Considering that there is already some works on incorporating auxiliary information into the compatibility calculation (as shown in Table 1), it would be beneficial to evaluate the performance of this function when it is only used in AAFM.

3. This paper adopts a three-stage training scheme, what is the effect of each stage on the performance?

**Questions:**

Please refer to the weaknesses section.

---

### Official Review · Reviewer_VxpP · 2024-11-03

**Soundness:** 2
**Presentation:** 3
**Contribution:** 2
**Rating:** 5
**Confidence:** 4

**Summary:**

The paper proposes the Instance-Conditioned Adaptation Model (ICAM) to enhance the generalization performance of reinforcement learning-based neural combinatorial optimization (NCO) models, particularly for vehicle routing problems (VRPs). The model leverages a novel instance-conditioned adaptation function and replaces the traditional multi-head attention (MHA) with the Adaptation Attention Free Module (AAFM). While ICAM demonstrates competitive empirical results and scalability, several aspects remain insufficiently explored.

**Strengths:**

1. The proposed ICAM shows promising results on VRPs with larger scales, successfully extending the capability of RL-based NCO methods to tackle instances with up to 5,000 nodes, which contributes positively to the ongoing effort of solving large-scale combinatorial optimization problems.
2. Introducing an adaptation function to incorporate instance-specific information into the optimization process is an interesting idea that can potentially improve the solution quality for diverse problem scales.

**Weaknesses:**

1. While the paper presents a new model named ICAM, its innovative contributions appear limited in scope. Specifically, the proposed modifications primarily target the adaptation function and attention mechanism, and these incremental changes might not be sufficient to demonstrate a significant advancement over existing RL-based NCO methods.
2. The term "comprehensive instance-conditioned information" is not clearly defined. The paper states that using only "node-to-node distances or scale information is insufficient." However, in the proposed adaptation function, these are the only two types of information used. There is no addition of new forms of instance-specific information beyond these, which seems to contradict the paper's claims of comprehensive information usage.
3. The design choices for both the adaptation function and the attention mechanism lack rigorous theoretical analysis.
In the "Results on Benchmark Dataset" section, the performance on TSPLIB is not particularly competitive compared to some of the state-of-the-art methods. The reasons for this subpar performance should be further analyzed.
4. The writing, particularly in the contribution section and the motivation (Section 2.1), includes several redundant explanations. A more concise presentation of the contributions and key ideas would improve readability.

**Questions:**

1. ICAM is presented as an instance-conditioned model, the datasets used for validation, however, consist only of uniformly distributed nodes. Demonstrating its performance on datasets with diverse distributions would make the 'instance-conditioned' claim more convincing.
2. As an "instance-conditioned" model, ICAM should be capable of capturing deeper feature relationships. How does the paper reflect the capture of instance features?
3. Although the authors state that AAFM has a lower computational complexity compared to MHA, the paper does not provide a quantitative analysis of the model’s overall complexity and memory usage.

---

### Official Review · Reviewer_Rd2k · 2024-11-04

**Soundness:** 3
**Presentation:** 3
**Contribution:** 2
**Rating:** 5
**Confidence:** 4

**Summary:**

This paper presents ICAM, a simple, efficient instance-conditioned adaption function to improve the generalization performance of existing NCO models for solving routing problems. They further propose a low-complexity instance-conditioned adaption module to generate better solutions for different scale instances. The authors claim and demonstrate that it achieves state-of-the-art performance among all RL-based constructive methods for TSPs, ATSPs (up to 1000 nodes), and CVRPs (up to 5000 nodes).

**Strengths:**

- The overall presentation and writing are good.
- The reported results show that ICAM is capable of obtaining promising results with a reasonable inference time in solving TSP, CVRP, and ATSP.
- This method can improve the performance of RL-based constructive neural solver, i.e., POMO, for TSP and ATSP up to 1000 nodes and 5000 nodes on CVRP instances.

**Weaknesses:**

- In the Introduction (lines 84-91), the authors state that all the previous methods that utilize the node-to-node distances to bias the output score in the decoding phase or refine the information via a complex policy fail to achieve satisfactory generalization performance on large-scale instances. This statement appears inaccurate, as these methods, e.g., ELG [1], are known to achieve state-of-the-art generalization performance on large-scale TSP and CVRP instances.
- In the adaption function, the learnable parameter $\alpha$ plays important role in this function. This paper lacks clarity on how this parameter is learned during training, and a detailed analysis is necessary to assess its impact.
- In experiments, this paper uses a three-stage training scheme for the proposed model. Stages 1 and 2 are not new and are similar to the training techniques in DAR [2]. In stage 3, there are two additional parameters, i.e., $\beta$ and $k$. The authors claimed that there is no significant performance variation among different models with various $k$ values. This raises a concern about the impact of the best $k$ trajectories among all $N$ trajectories on the performance of the model.
- Furthermore, additional training may make the results unfair to compare the performance with other existing baseline models. The results of baseline models with expanding training should be reported and compared.
- The author only reported the results of baseline models, e.g., LEHD [3] and BQ [4], with greedy search inference for TSP and CVRP with scale <=1000 (Table 3). However, these models achieve their best performance with the RRC strategy and beam search, respectively. According to these works, the results obtained from these baselines can surpass the ICAM results. The authors should report these results for comprehensive comparisons.
- With scales >1000 (Table 5), the results of the baseline ELG are lacking, while ELG demonstrated superior performance on large-scale CVRP instances up to 7000 nodes. However, the best results obtained by ICAM do not seem much better than those obtained by LEHD with greedy inference. It makes the results of ICAM on CVRP instances with a scale range from 1000 to 5000 not persuasive.
- In the results on the benchmark dataset, this paper just evaluates the performance on a small-scale dataset of CVRPLIB Set-X (scale <=1000). The performance on larger scale datasets, such as CVRPLIB Set-XXL with larger scales (1000 to 7000 nodes) should be evaluated and analyzed.
- The same concern about results for TSPLIB, the author only reported the results on instances with scale <=1000, which limits the large-scale generalization performance on larger scales (>1000) in the TSPLIB. This evaluation is important to verify the good generalization of the proposed method on large-scale Routing problems, e.g., TSP and CVRP.
- In the ablation study on the effects of larger training scales, the authors only report the results for TSP and not for CVRP. The proposed varying scale training on CVRP should be carefully designed and investigated because CVRP has more complex features than TSP, such as varying demand at each node, which may destabilize the learning model when exposed to multiple sizes and demand distributions simultaneously.
- No code is available.


**References:**
[1] Chengrui Gao, Haopu Shang, Ke Xue, Dong Li, and Chao Qian. Towards generalizable neural solvers for vehicle routing problems via ensemble with transferrable local policy. In International Joint Conference on Artificial Intelligence, 2024.

[2] Yang Wang, Ya-Hui Jia, Wei-Neng Chen, and Yi Mei. Distance-aware attention reshaping: En- hance generalization of neural solver for large-scale vehicle routing problems. arXiv preprint arXiv:2401.06979, 2024.

[3] Neural combinatorial optimization with heavy decoder: Toward large scale generalization. In Thirty-seventh Conference on Neural Information Processing Systems, 2023.

[4] Darko Drakulic, Sofia Michel, Florian Mai, Arnaud Sors, and Jean-Marc Andreoli. Bq-nco: Bisim- ulation quotienting for efficient neural combinatorial optimization. In Thirty-seventh Conference on Neural Information Processing Systems, 2023.

**Questions:**

The questions are mentioned in the weaknesses. Please refer to the weakness.

---

### Meta-Review · Area_Chair_iUBd · 2024-12-20

**Metareview:**

This paper proposed a instance conditioned adaptation method to enhance the generalization ability of neural vehicle routing models. The proposed method has the strengths of being simple but effective, achieving good empirical performance. However, as mentiond by most reviewers, the technical novelty is not sufficient enough to warrant acceptance to ICLR. It largely relies on existing lightweight attention techniques, and the proposed instance-conditioned adaptation function is heuristic without much theoretical foundation. In addition, the varying-size and three-stage training scheme plays an important role in performance improvement, which seems irrelavent to the instance-conditioned theme. Overall, this is a boarderline paper, and the reasons of rejection outweight that of acceptance.

**Additional Comments On Reviewer Discussion:**

Reviewers raised concerns regarding novelty of the proposed method and fairness of the experimental evaluation. During rebuttal, authors provided quite a few new results. However, many of the concerns still remain. While one reviewer raised score, the overall evaluation is still boarderline. I tend to agree that the technical contribution is somewhat limited, and therefore lean towards rejection.

---

### Decision · Program_Chairs · 2025-01-22

Reject